

# The nitrogen pendulum in Sandusky Bay, Lake Erie: Oscillations between strong and weak export and implications for harmful algal blooms

Kateri R. Salk[1], George S. Bullerjahn[2], Robert Michael L. McKay[2], Justin D. Chaffin[3], Nathaniel E. Ostrom[1]

[1]Department of Integrative Biology, Michigan State University, East Lansing, MI 48824 United States
[2]Department of Biological Sciences, Bowling Green State University, Bowling Green, OH 43403 United States
[3]F.T. Stone Laboratory Ohio Sea Grant, The Ohio State University, Put-in-Bay, OH 43456 United States

*Correspondence to*: Kateri R. Salk (krsalkgu@uwaterloo.ca)

**Abstract.** Recent global water quality crises point to an urgent need for greater understanding of cyanobacterial harmful algal blooms (cHABs) and their drivers. Nearshore areas of Lake Erie such as Sandusky Bay may become seasonally limited by nitrogen (N) and are characterized by distinct cHAB compositions (i.e., *Planktothrix* over *Microcystis*). This study investigated phytoplankton N uptake pathways, determined drivers of N depletion, and characterized the N budget in Sandusky Bay. Nitrate ($NO_3^-$) and ammonium ($NH_4^+$) uptake, N fixation, and N removal processes were quantified by stable isotopic approaches. Dissimilatory N uptake was a relatively modest N sink, with denitrification, anammox, and $N_2O$ production accounting for 84, 14, and 2 % of N removal, respectively. Phytoplankton assimilation was the dominant N uptake mechanism, and $NO_3^-$ uptake rates were higher than $NH_4^+$ uptake rates. Riverine DIN loading was sometimes insufficient to meet assimilatory and dissimilatory demands, but N fixation alleviated this deficit. N fixation made up 23.7-85.4 % of total phytoplankton N acquisition and indirectly supports *Planktothrix* blooms. However, N fixation rates were surprisingly uncorrelated with $NO_3^-$ or $NH_4^+$ concentrations. Owing to temporal separation in sources and sinks of N to Lake Erie, Sandusky Bay pendulums between acting as a strong and weak source of downstream N loading to Lake Erie. Estuarine systems such as Sandusky Bay are mediators of downstream N loading, but climate change-induced increases in precipitation and N loading will likely intensify the swings of the N pendulum in favor of N export.

## 1 Introduction

Harmful algal blooms (HABs) are increasing in frequency on a global scale and are stimulated by excessive nutrient loading to aquatic systems (Bricker et al., 2008; Heisler et al., 2008). Lake Erie, in particular, has been subject to increased incidence and expansion of cyanobacterial HABs (cHABs) in recent years (Michalak et al., 2013; Ho and Michalak, 2015, Bullerjahn et al., 2016). These blooms are dominated by cyanobacteria that accumulate the powerful hepatotoxin, microcystin (Carmichael and Boyer, 2016). A more nuanced understanding of the drivers of cHABs, including nutrient cycling, will allow for better prediction and management of blooms in Lake Erie and other ecosystems.





Phytoplankton biomass and cHABs in Lake Erie have historically been correlated with P loading from river inflows (Kane et al., 2014; Kim et al., 2014). Calls to control eutrophication in Lake Erie have proposed targets for reduced P loading but have largely ignored N (Scavia et al., 2014). However, there is a growing dialogue surrounding the dual management of N

and P in lacustrine systems (Gobler at al., 2016; Paerl et al., 2016), particularly as co-limitation of phytoplankton growth by both N and P has been demonstrated in the late summer in Lake Erie (Moon and Carrick, 2007; North et al., 2007; Chaffin et al., 2013; Steffen et al., 2014a). This seasonal N deficiency is consistent with reduced watershed loading of N into the lake's western basin over the past two decades (Stow et al., 2015) combined with active dissimilatory sinks for nitrate (Small et al., 2016). N concentration and speciation also influence toxin production by cHABs (Horst et al., 2014; Monchamp et al., 2014;

Davis et al., 2015). Delineating the role of N in controlling cHABs, therefore, requires investigation of spatial and temporal variation in multiple species of N.

Because the Lake Erie catchment is highly agricultural, the majority of bioavailable N in the lake is derived from farming practices and delivered to the lake via river inflows (Robertson and Saad, 2011; Stow et al., 2015). Once in the lake,

dissolved inorganic N (DIN) and dissolved organic N (DON) are subject to consumption by competing biological processes. Phytoplankton, including cHAB taxa, commonly take up DIN and DON in the form of ammonium ($NH_4^+$), nitrate ($NO_3^-$), and urea (Davis et al., 2015). N fixation, an alternate source of bioavailable N, is expected to occur when DIN is scarce. N fixation activity has been inferred in Lake Erie (MacGregor et al., 2001; Monchamp et al., 2014; Steffen et al., 2014a; Davis et al., 2015) but not quantified for several decades (Howard et al., 1970). Dissimilatory microbial processes may also

consume DIN in sediments, representing pathways of permanent N removal whereby N leaves the system as $N_2$. Denitrification is expected to be the dominant microbial N removal pathway in freshwaters (Seitzinger et al., 2006), although anammox, a competing pathway, has not been extensively studied in freshwater systems (Yoshinaga et al., 2011; Zhu et al., 2013). Nitrous oxide ($N_2O$), a potent greenhouse gas, is a byproduct of denitrification and nitrification (Wrage et al., 2001) and is a detrimental consequence of microbial N removal.

Whereas the colonial cyanobacterium *Microcystis* dominates the cHAB community in offshore regions of western Lake Erie, filamentous *Planktothrix* has been shown to persist in N-limited bays and tributaries (Conroy et al., 2007; Kutovaya et al., 2012; Davis et al., 2015). With few exceptions (Pancrace et al., 2017), both cHAB taxa are incapable of N fixation and require dissolved forms of N for growth. *Planktothrix* is a superior scavenger for N (Conroy et al., 2007) and responds

strongly to additions of DIN (Donald et al., 2011; 2013). Under N limitation, *Planktothrix* may thus be able to outcompete other phytoplankton for small N inputs. *Planktothrix* is also particularly well-adapted to low irradiances; suspended sediment that rapidly attenuates light is often found in *Planktothrix*-dominated waters (Scheffer et al., 1997; Havens et al., 2003; Oberhaus et al., 2005). Additionally, *Planktothrix* can tolerate the widest temperature range among major bloom-forming cyanobacteria, including *Microcystis, Aphanizomenon*, and *Dolichospermum (Anabaena)* (Foy et al., 1976; Post et al., 1985).




Due to these factors, the persistence of *Planktothrix* in nearshore zones likely operates under a fundamentally different paradigm than offshore *Microcystis* blooms, and mitigation may require attention to the distinct biogeochemical functioning of these genera in the nearshore vs. offshore.

Sandusky Bay, an estuary on the southern shore of Lake Erie, serves as an ideal location in which to examine the relationship between N cycling and cHABs. This system is hypereutrophic (Ostrom et al., 2005), with the cyanobacterium *Planktothrix* dominating phytoplankton biomass from May to October and N fixing phytoplankton making up only a small portion of biomass (Davis et al., 2015). Sandusky Bay experiences large fluctuations in $NO_3^-$ concentrations and dissolved N:P ratios throughout the summer (Davis et al., 2015; Conroy et al., 2017), suggesting that the dynamic N cycling may

influence cHAB formation in this system. Indeed, growth of *Planktothrix* in Sandusky Bay is stimulated by additions of $NO_3^-$, $NH_4^+$, and urea, indicating that phytoplankton growth is seasonally limited or co-limited by N (Chaffin and Bridgeman, 2014; Davis et al., 2015). Evaluating the mechanisms that promote the persistence of *Planktothrix* in this system will benefit from an examination of N removal processes and inputs from N fixation that directly influence the availability of DIN. A thorough understanding of these processes will also inform the capacity for Sandusky Bay to act as a nutrient sink,

which may curtail the formation of cHABs in offshore regions of Lake Erie.

The objectives of this study, conducted in Sandusky Bay, Lake Erie, were to (1) investigate the pathways by which the *Planktothrix*-dominated phytoplankton community acquires N, (2) determine the factors driving N depletion, namely microbial N removal processes and hydrology, and (3) evaluate Sandusky Bay as a Great Lakes estuary in which N cycling

affects cHABs and N loading to Lake Erie. These objectives provide insight into the Grand Challenges of Great Lakes research by addressing variability in ecosystem processes, anthropogenic nutrient forcing and potential for reversibility, and response of this ecosystem to climate change (Sterner et al., 2017).

## 2 Methods

### 2.1 Field Sampling

Sampling took place between May and October in 2015 and 2016 in partnership with the Ohio Department of Natural Resources (ODNR). Water column nutrient and chlorophyll (chl) *a* samples were collected weekly in 2015 and approximately biweekly in 2016. Samples for DIN uptake assays (2016), N fixation assays (2015 and 2016), and microbial N removal assays (2015) were collected monthly. Six stations were sampled: two stations in the inner portion of Sandusky Bay (ODNR4, ODNR6), three stations in the outer portion of Sandusky Bay (ODNR2, ODNR1, and Environment Canada

1163, hereafter 1163), and one station directly outside Sandusky Bay in the western basin of Lake Erie (Bells; Fig. 1). Tributary discharge data from the primary water source to Sandusky Bay, the Sandusky River, were obtained from the USGS stream monitoring station near Fremont, OH (site 04198000; Fig. 1). Hydraulic residence time in Sandusky Bay was




estimated by dividing the Bay volume (1.6 - 2.6 m mean depth, 162 km$^2$ area; Richards and Baker, 1985) by the Sandusky River discharge rate. $NO_3^-$ concentrations in the Sandusky River near the USGS monitoring site were provided by the Heidelberg Tributary Loading Program maintained by the National Center for Water Quality Research (NCWQR) at Heidelberg University (Richards et al., 2010).

At each sampling location, water column physical and chemical parameters (pH, conductivity, temperature, dissolved oxygen) were measured using a YSI 600QS sonde (YSI Inc., Yellow Springs, OH). Water samples were collected by Van Dorn bottle at 1 m depth for analysis of $NO_3^-$, $NH_4^+$, phosphate ($PO_4^{3-}$), and chl *a* concentrations. Samples for dissolved nutrient analysis were filtered immediately upon collection (0.2 μm), kept on ice, and frozen upon return to the lab. Samples

for determination of chl *a* concentrations were collected on 0.2 μm polycarbonate membrane filters and frozen. Station 1163 has an extensive monitoring history by Environment and Climate Change Canada and was chosen for additional water and sediment assays. An additional 20 L carboy was filled with surface water from station 1163 for N uptake assays and sediment incubations. Sediment cores for the evaluation of microbial N removal rates (denitrification, anammox, and $N_2O$ production) by the isotope pairing technique (IPT) were collected at station 1163 using a modified piston corer as described

by Smit and Steinman (2015). Intact sediment cores were collected in polycarbonate tubes (7 cm i.d.) to a depth of 25 cm. Water (1 cm) was maintained overlying the sediment during transport to preserve redox gradients.

**2.2 Nutrient and Chlorophyll *a* Analyses**

Concentrations of $NO_3^-$ + $NO_2^-$, $NO_2^-$, $NH_4^+$, and $PO_4^{3-}$ were measured on field-filtered sample water using standard U.S. EPA methods (353.1, 353.2, 350.1, and 365.1, respectively) on a SEAL Analytical QuAAttro continuous segmented flow

analyzer (SEAL Analytical Inc., Mequon, WI). $NO_3^-$ concentration was determined as the difference between $NO_3^-$ + $NO_2^-$ and $NO_2^-$. Seven known concentration standard solutions (including 0) were used for the standard curve ($R^2 > 0.999$), and every-tenth sample was spiked with a known amount of analyte to ensure high accuracy and precision throughout the analysis (> 95 % recovery). Samples with concentrations exceeding the highest standard were diluted and reanalyzed. Values were averaged over two or three replicates. Method detection limits were 0.165, 0.044, 0.558, and 0.044 μmol L$^{-1}$ for $NO_3^-$ +

$NO_2^-$, $NO_2^-$, $NH_4^+$, and $PO_4^{3-}$, respectively. Extractive chl *a* concentration was measured following Welschmeyer (1994). Filters containing phytoplankton seston were extracted with 90% aqueous acetone overnight at -20 °C followed by measurement of the clarified extract by fluorometry (model TD-700, Turner Designs, Sunnyvale, CA).

**2.3 Sediment Microbial N Removal**

Upon return to the lab, water from station 1163 was gently added to cores to a depth of 20 cm. Cores were pre-incubated for

12 h in the dark at *in situ* temperature under gentle aeration to maintain oxic conditions in the overlying water. Following pre-incubation of sediment cores, a sample was taken from the overlying water for DIN concentration analysis ($NO_3^-$, $NO_2^-$ and $NH_4^+$), filtered through a precombusted GF/F filter and frozen until analysis. $^{15}N$-$NO_3^-$ (100 μmol L$^{-1}$) was then added to





the overlying water in each core. Cores were then capped and statically incubated under gentle stirring throughout the duration of the experiment. An initial equilibration period was employed to allow homogenization of $NO_3^-$ between the overlying water and the $NO_3^-$ reduction zone in the sediment porewater (Dalsgaard et al., 2000). Cores were sacrificed in triplicate or quadruplicate at intervals of 0, 3 or 4, and 6 h, during which time oxic conditions were maintained in the overlying water. Dissolved $O_2$ in the overlying water was monitored to evaluate the maintenance of oxic conditions throughout the incubation using a YSI 600QS sonde. A final sample for DIN analysis was collected when each core was sacrificed and processed as described above.

Samples for the determination of $\delta^{15}N_2$ were collected according to Hamilton and Ostrom (2007); briefly, dissolved gases were equilibrated with a He atmosphere, and the headspace was transferred into a pre-evacuated 12 mL Exetainer (Labco Ltd, Lampeter, Ceredigion, UK). Samples for analysis of dissolved $N_2$ concentrations were siphoned into 12 mL Exetainers to overflowing and amended with 200 µL of saturated $ZnCl_2$ solution to halt biological activity. All Exetainers were stored underwater at room temperature to minimize diffusion of atmospheric $N_2$ during storage. Samples for analysis of the $\delta^{15}N_2O$ and $N_2O$ concentrations were siphoned into 250 and 60 mL serum bottles, respectively, to overflowing and sealed without a headspace with a butyl rubber septum. Biological activity was halted by adding saturated $HgCl_2$ solution to a final concentration of 0.4 % by volume.

Prior to $N_2O$ concentration analysis, a headspace of 20 mL He was introduced in each 60 mL bottle, maintaining atmospheric pressure with a vent needle. Serum bottles were allowed to equilibrate under gentle shaking for at least 12 h prior to analysis. The headspace was then analyzed by GC-ECD (Shimadzu Greenhouse Gas Analyzer GC-2014, Shimadzu Scientific Instruments, Columbia, MD) for $N_2O$ concentration. The dissolved concentration was calculated based on the headspace equilibrium concentration (Hamilton and Ostrom, 2007).

The isotopic composition of $N_2O$ was analyzed upon introduction of sample water into an enclosed 0.75 L glass vessel that was previously purged of atmospheric air using a gentle flow of He. Dissolved gases were subsequently stripped from the water by sparging the sample with He (Sansone et al., 1997), which carried sample gases into a Trace Gas sample introduction system interfaced to an Isoprime isotope ratio mass spectrometer (IRMS; Elementar Americas, Inc., Mount Laurel, NJ). The relative abundance of a stable isotope within a particular material or reservoir is reported in standard delta notation:

$$\delta = \frac{R_{sam} - R_{std}}{R_{std}} \times 1000 \tag{1}$$

where $R_{sam}$ is the isotope ratio of the sample, $R_{std}$ is the isotope ratio of the standard, and $\delta$ is reported as per mil (‰).



Concentrations of dissolved $N_2$ were analyzed by membrane inlet mass spectrometry (MIMS; Kana et al., 1994). The isotopic composition of $N_2$ was analyzed by introducing the sample to an evacuated 800 µL sampling loop and then onto a packed molecular sieve (5 Å) column (Alltech, Inc., Deerfield, IL) using He carrier gas within a gas chromatograph (HP-5980, Hewlett Packard, Ramsey, MN) interfaced to an Isoprime IRMS. Analytical reproducibility of standards was 0.3 ‰.

Denitrification, anammox, and $N_2O$ production rates were calculated by the IPT. Calculations were derived from the $IPT_{anaN2O}$ (Hsu and Kao, 2013), which builds on the R-IPT (Risgaard-Petersen et al., 2003) by enabling quantification of $N_2O$ production simultaneously with denitrification and anammox. Briefly, $N_2$ production by denitrification ($D_{14\text{-}N2}$) and anammox ($A_{14}$) were calculated as:

$$D_{14\text{-}N_2} = (r_{14\text{-}N_2O}+1) \times 2r_{14\text{-}N_2O} \times P_{30} \tag{2}$$

and

$$A_{14} = 2r_{14\text{-}N_2O} \times (P_{29}\text{-}2r_{14\text{-}N_2O} \times P_{30}) \tag{3}$$

where $r_{14\text{-}N2O}$ is the ratio of $^{14}N$ to $^{15}N$ in $N_2O$, $P_{29}$ is the production of $^{29}N_2$, and $P_{30}$ is the production of $^{30}N_2$. $N_2O$ production ($D_{14\text{-}N2O}$) was calculated as:

$$D_{14\text{-}N_2O} = r_{14\text{-}N_2O} \times (2P_{46}+ P_{45}) \tag{4}$$

where $P_{45}$ and $P_{46}$ are the production of $^{45}N_2O$ and $^{46}N_2O$, respectively.

## 2.4 Phytoplankton N Uptake

DIN uptake assays were conducted by adding $^{15}NO_3^-$ or $^{15}NH_4^+$ to a serum bottle containing site water to a target concentration of 10 % of ambient $NO_3^-$ or $NH_4^+$ concentration, respectively. Each assay for a single site was run in triplicate.

Bottles were placed in an incubator at *in situ* light intensity, light:dark cycle, and temperature conditions for 24 h.

N fixation assays were conducted by the dissolution method, which involves the addition of $^{15}N_2$-equilibrated water to a water sample rather than a $^{15}N_2$ bubble (Großkopf et al., 2012). Preparation of $^{15}N_2$-equilibrated water involved sparging water in a serum bottle equipped with a butyl rubber septum with He to remove ambient $N_2$, followed by injection of $^{15}N_2$

(98 % atom fraction, Sigma-Aldrich Lot #MBBB0968V) while maintaining atmospheric pressure with a vent needle. Water from station 1163 was transferred into 1.18 L serum bottles and amended with $^{15}N_2$-equilibrated water to a final dissolved atom fraction of 1.14-2.33 %. Each assay for a single site was run in triplicate. Bottles were incubated at *in situ* light and temperature conditions for 24 h.

To evaluate the potential contamination of $^{15}N_2$ gas with $^{15}NO_3^-$ or $^{15}NH_4^+$ (Dabundo et al., 2014), a mass scan of the isotopically enriched gas was performed by IRMS. The mass scan revealed that potential impurities made up < 1 % of the





enriched gas. The maximum contamination reported by Dabundo et al. (2014), though not detected, could have made up less than 5 % of measured N fixation rates if all available contaminating $^{15}NO_3^-$ and $^{15}NH_4^+$ was assimilated.

When DIN uptake and N fixation incubations were complete, the samples were vacuum filtered through precombusted GF/F

filters. Filters were then dried at 60° C, acidified with 10 % HCl to remove carbonates, and dried again. The concentration and isotopic composition of particulate organic matter (POM) from N fixation assays was analyzed by scraping the contents of dried and acidified filters into tin cups and introducing samples to an elemental analyzer interfaced to an Isoprime IRMS. Analytical reproducibility of standards was 0.2 ‰.

DIN uptake rates were calculated according to Dugdale and Wilkerson (1986). The transport rate, or N uptake rate ($\rho$; μmol N $L^{-1}$ $h^{-1}$), was calculated as:

$$\rho_t = \frac{^{15}N_{xs}}{(^{15}N_{enr} - {}^{15}N_{na}) \times T} \times PON_t \qquad (5)$$

where $^{15}N_{xs}$ is the atom percent $^{15}N$ excess in the POM sample, $^{15}N_{enr}$ is the isotope fraction of $^{15}N$ in the $NO_3^-$ or $NH_4^+$ pool, and $^{15}N_{na}$ is the isotope fraction of $^{15}N$ in the natural abundance POM. T is the time in hours, and $PON_t$ is the concentration

of N in POM (μmol N $L^{-1}$) at the end of the incubation.

N fixation rates were calculated according to Montoya et al. (1996). The transport rate, or N fixation rate ($\rho$; μmol N $L^{-1}$ $h^{-1}$), was calculated as:

$$\rho_t = \frac{A_{PN_f} - A_{PN_0}}{(A_{N_2} - A_{PN_0}) \times T} \times PON_t \qquad (6)$$

where $A_{PN0}$ and $A_{PNf}$ are the isotope fractions of $^{15}N$ in the POM at the start and end of the incubation, respectively, and $A_{N2}$ is the isotope fraction of $^{15}N$ in the labeled $N_2$ pool. Areal N fixation rates were calculated by scaling volumetric rates over the depth of the photic zone in proportion to light attenuation, under the assumption that water column N fixation is light-dependent (Scott and Grantz, 2013).

## 2.5 N Budget

To enable direct comparisons with water column N cycling rates, sediment N removal rates were converted to volumetric rates by multiplying areal rates by the depth of the water column (3.2 m). Hourly N cycling rates were converted to daily rates by multiplying by the hours of daylight (11.25–15 h) for phytoplankton N uptake and N fixation and by 24 h for sediment N removal processes (denitrification, anammox, and $N_2O$ production). These calculations assumed that phytoplankton N uptake and N fixation were light-dependent but sediment N removal was light-independent.





In order to generate a preliminary budget of the magnitude of N sources and sinks in Sandusky Bay, total daily loads of each process for the entire system (kg d$^{-1}$) were calculated by scaling up volumetric measurements. The total NO$_3^-$ load to Sandusky Bay for each sampling date was estimated by multiplying Sandusky River discharge by the NO$_3^-$ concentration in the Sandusky River. Total Kjeldahl N (TKN) was also measured by the NCWQR alongside NO$_3^-$, but TKN was omitted

from the N loading budget because TKN makes up a small portion of the total N load in the Sandusky River and includes a pool of refractory N that was not relevant to the study objectives. NH$_4^+$ data were not available for the Sandusky River, so the total NH$_4^+$ load for each sampling date was estimated by multiplying discharge by the NH$_4^+$ concentration at the Sandusky Bay site nearest the Sandusky River outlet (ODNR4). The error associated with this assumption is expected to be low, as NH$_4^+$ represented a minor fraction of DIN. The total DIN load to Sandusky Bay was calculated as the sum of NO$_3^-$

and NH$_4^+$ loading. System-wide N cycling rates (kg d$^{-1}$) were estimated by multiplying volumetric daily rates by the volume of Sandusky Bay (0.26 km$^3$). While point estimates of N cycling processes are not necessarily representative of the entire system, these calculations made it possible to compare the relative magnitudes of N sources, assimilatory uptake processes, and dissimilatory sinks in Sandusky Bay.

**2.6 Statistics**

Statistical modeling was carried out in R (version 3.2.4). Correlations between (1) nutrient concentrations and discharge and (2) N cycling rates and DIN concentrations were analyzed by linear regression. Pairwise differences between NO$_3^-$ and NH$_4^+$ uptake rates were analyzed by *t*-test. Differences in rates of denitrification, anammox, and N$_2$O production by date were analyzed by one-way ANOVA, and subsequent pairwise differences were determined by Tukey's HSD post-hoc test. Differences in NO$_3^-$ uptake, NH$_4^+$ uptake, and N fixation by date and station were analyzed by two-way interaction effects

ANOVA, and subsequent pairwise differences were determined by Tukey's HSD post-hoc test. In all cases, date was treated as a fixed effect, and temporal autocorrelation was avoided owing to the spacing of sampling dates approximately one month apart.

**3 Results**

Discharge from the Sandusky River was highly episodic in both years (Fig. 2k, 2l) and dominated by significant rain events.

Owing to more frequent and intense precipitation in 2015, peak discharge was an order of magnitude higher than in 2016. NO$_3^-$ and PO$_4^{3-}$ concentrations were positively correlated with discharge (NO$_3^-$: df = 135, R$^2$ = 0.26, *p* < 0.0001; PO$_4^{3-}$: df = 135, R$^2$ = 0.05, *p* < 0.01), but NH$_4^+$ concentrations were not correlated with discharge (df = 135, R$^2$ = 0.01, *p* = 0.08). The shortest hydraulic residence time in 2015 was 8 days during peak discharge but in 2016 was 82 days owing to lower peak discharge. In both years, as discharge decreased in the late summer and early fall, hydraulic residence time increased to

several months.





$NO_3^-$ concentrations ranged from below detection ($< 0.1$ µmol $L^{-1}$) to a maximum of $> 650$ µmol $L^{-1}$ in 2015 and 65 µmol $L^{-1}$ in 2016 (Fig. 2a, 2b). Temporal patterns in $NO_3^-$ concentration in the bay followed those in the Sandusky River. In both years, the greatest $NO_3^-$ concentrations were observed in June and July followed by a decline in August that continued through October. The magnitude of these shifts was greater for the inner bay stations (ODNR4, ODNR6) than the outer bay

stations (ODNR1, ODNR2, 1163). $NO_3^-$ concentrations at the nearshore Lake Erie station (Bells) displayed similar temporal patterns to those in the bay in 2015, whereas $NO_3^-$ concentrations at Bells were consistently higher than those in the bay in 2016. $NH_4^+$ concentrations ranged from below detection ($< 0.5$ µmol $L^{-1}$) to 17.5 µmol $L^{-1}$ across sites during the sampling period and were generally lower in 2016 than 2015 (Fig. 2c, 2d). The greatest $NH_4^+$ concentrations were observed in the inner bay, and these spikes occurred episodically throughout the sampling season. $PO_4^{3-}$ concentrations ranged from below

detection ($< 0.04$ µmol $L^{-1}$) to 4.25 µmol $L^{-1}$ across sites during the sampling period, and the highest concentrations were observed in the inner bay (Fig. 2e, 2f). The molar ratio of DIN to dissolved inorganic P (N:P; $NO_3^-$ + $NH_4^+$ to $PO_4^{3-}$) was highly variable throughout the sampling period in 2015 and 2016, ranging from over 10,000 to below Redfield stoichiometry (16:1) (Fig. 2g, 2h). In general, high N:P values were observed earlier in the season, and low N:P values were observed later in the season as $NO_3^-$ concentrations declined. Chl $a$ concentrations ranged from 3.5 to nearly 150 µg $L^{-1}$ across sites

throughout the sampling period in 2015 and 2016 (Fig. 2i, 2j). Maximum chl $a$ concentrations in both years occurred in late August to early September, approximately one month after the peak in $NO_3^-$ and $PO_4^{3-}$ concentrations.

Sediment N removal processes were active across the sampling period at station 1163 in 2015 (Fig. 3). Denitrification rates ranged from 10.02-64.81 µmol N $m^{-2}$ $h^{-1}$ over all sampled dates, decreasing over time coincident with declines in water

column $NO_3^-$ (ANOVA, $F_{3,24} = 6.53$, $p < 0.01$). Anammox activity was detected on all sampling dates but not in all replicate sediment cores. Anammox rates ranged from 0.52 - 8.10 µmol N $m^{-2}$ $h^{-1}$ across sampled dates, displaying no clear temporal trend (ANOVA, $F_{3,24} = 2.60$, $p = 0.08$). The majority of measured anammox rates were less than 7 µmol N $m^{-2}$ $h^{-1}$, with the exception of two cores on July 27 that displayed elevated anammox rates of 15.15 and 30.75 µmol N $m^{-2}$ $h^{-1}$. $N_2O$ production rates at station 1163 ranged from 0.09-2.34 µmol N $m^{-2}$ $h^{-1}$ across sampled dates, decreasing over time coincident with

declines in water column $NO_3^-$ (ANOVA, $F_{3,24} = 6.85$, $p < 0.01$). Denitrification, anammox, and $N_2O$ production made up an average of 84 %, 14 %, and 2 % of total N removal, respectively.

$NO_3^-$ and $NH_4^+$ uptake were active in the inner bay, outer bay, and nearshore Lake Erie throughout the sampling period in 2016. $NO_3^-$ uptake rates ranged from 0.01-1.92 µmol N $L^{-1}$ $h^{-1}$ (Fig. 4a), and $NH_4^+$ uptake rates ranged from 0.001-0.11 µmol

N $L^{-1}$ $h^{-1}$ (Fig. 4b). There was a significant interaction between the effects of date and location on $NO_3^-$ uptake (ANOVA, $F_{6,24} = 133.14$, $p < 0.0001$) and $NH_4^+$ uptake (ANOVA, $F_{6,24} = 19.43$, $p < 0.0001$), and rates across dates and sites clustered into significant groupings (marked with letters in Fig. 4a, 4b). Overall, $NO_3^-$ and $NH_4^+$ uptake rates were higher within the bay than in the nearshore Lake Erie station. Rates of $NO_3^-$ uptake were significantly higher than rates of $NH_4^+$ uptake ($t$-test, df = 35, T = 2.41, $p = 0.02$). As a result of ambient concentrations being lower than anticipated, there were several instances

when the $^{15}$N-labeled fraction of $NO_3^-$ or $NH_4^+$ exceeded 10 % (marked with ^ in Fig. 4a, 4b). However, the elevation of substrate concentration was not associated with unusually high uptake rates in comparison with other dates at the same site, and these observations were retained in the dataset.

Water column N fixation was active throughout the sampling period in 2015 and 2016, with rates ranging from 0.06-2.16 $\mu$mol N $L^{-1}$ $h^{-1}$ (Fig. 4c). N fixation rates varied significantly by date (ANOVA, $F_{7,25} = 156.33$, $p < 0.0001$), and rates across dates fell into significant groupings (marked with letters in Fig. 4c). The highest observed rates of N fixation occurred in late July and late August of 2015. The lowest observed rates of N fixation occurred in late July and late August of 2016. Areal rates of N fixation ranged from 309.5-906.9 $\mu$mol N $m^{-2}$ $h^{-1}$ in 2015 and 0.2-187.8 $\mu$mol N $m^{-2}$ $h^{-1}$ in 2016.

DIN concentration was positively correlated with daily volumetric rates of denitrification, $N_2O$ production, and $NO_3^-$ uptake, explaining 64, 60, and 75 % of variance in mean rates, respectively (Fig. 5). Anammox rates were not correlated with DIN concentration due to the high variance observed among dates and replicate sediment cores (Fig. 5b). DIN concentration explained only 12 % of variance in mean $NH_4^+$ uptake rates, although $NH_4^+$ concentration alone explained 86 % of variance

and was positively correlated with $NH_4^+$ uptake. N fixation was not significantly correlated with DIN concentration (Fig. 5a). Ranges in daily volumetric rates of assimilatory processes ($NO_3^-$ uptake, $NH_4^+$ uptake, N fixation) were several orders of magnitude higher than dissimilatory processes (denitrification, anammox, $N_2O$ production), with the exception of $NH_4^+$ uptake and denitrification, which were within the same magnitude (Fig. 5).

Daily rates of DIN loading from the Sandusky River varied by six orders of magnitude and were tightly related to discharge (Table 1). N fixation was the greatest source of N to Sandusky Bay during periods of low DIN loading, but DIN loading exceeded N fixation as an N source in the early summer when Sandusky River discharge was high. $NO_3^-$ uptake was the dominant N uptake process in Sandusky Bay, outpacing $NH_4^+$ uptake and dissimilatory N removal processes (Table 1). On the basis of the total magnitude of ranges, sources and demands of N in this system are tipped in favor of a net source to

Lake Erie. The magnitude of the net source of N from Sandusky Bay to Lake Erie is at a minimum (nearly zero) when sources are at a minimum and greatest when sources are at a maximum (Table 1).

## 4 Discussion

### 4.1 Nutrient Stoichiometry

Sandusky Bay displays considerable seasonal variation in nutrient concentrations, molar dissolved N:P ratios, and chl $a$

concentrations, indicative of a system with dynamic changes in hydrology and biogeochemical activity. Maximum chl $a$ concentrations in both years (> 100 $\mu$g $L^{-1}$) were similar to other hypereutrophic systems (Zhang et al., 2011; Wheeler et al., 2012; Steffen et al., 2014b), as were large swings in $NO_3^-$ concentrations in 2015 (Xu et al., 2010; Steffen et al., 2014b;




McCarthy et al., 2016). Elevated nutrient concentrations were associated with high discharge events from the Sandusky River, demonstrating a strong watershed influence on Sandusky Bay. Indeed, the Sandusky River watershed comprises an area 30 times larger than the bay and delivers large nonpoint loads of N and P to its receiving waters (Robertson and Saad, 2011). Between the two study years, discharge from the Sandusky River varied substantially (tenfold higher in 2015), exhibiting large inter-annual variability in hydraulic residence time and nutrient concentrations.

As discharge from the Sandusky River decreased throughout the summer in both years, dissolved N:P ratios fell from a maximum of over 10,000 to below 16, the threshold for N limitation. The decline in N:P ratios is largely driven by decreases in $NO_3^-$ concentration, particularly in 2015, as the range in $PO_4^{3-}$ and $NH_4^+$ concentration was comparatively narrow. Consumption of $NO_3^-$ could be attributed to both assimilatory and dissimilatory $NO_3^-$ reduction. If phytoplankton were solely responsible for the decline in $NO_3^-$, nutrients would be expected to be drawn down in molar proportions of approximately 16N:1P (Sterner and Elser, 2002). However, N:P ratios in Sandusky Bay fall sharply throughout the summer, while $PO_4^{3-}$ concentrations are relatively constant by comparison. Although this trend could be influenced by luxury uptake of N by phytoplankton and internal P loading from sediments (Filbrun et al., 2013; McCarthy et al., 2016), the dramatic depletion in DIN compels consideration of microbial N removal processes as a major mechanism for N drawdown in Sandusky Bay.

### 4.2 N Removal Processes

Marked declines in N:P with time and occurrence of N:P ratios < 16 provide compelling evidence that microbial N removal processes (i.e., denitrification and/or anammox) consume appreciable quantities of $NO_3^-$ in Sandusky Bay. Sediment [15]N tracer incubations indicate the primary N removal mechanism in Sandusky Bay is denitrification, which comprised an average of 84 % of total N removal across sampling dates. Denitrification rates were positively correlated with DIN concentration, consistent with observations that N supply controls sediment denitrification capacity in estuaries, lakes, and continental shelves (Seitzinger et al., 2006). Denitrification rates in Sandusky Bay (10.02-64.81 µmol N m$^{-2}$ h$^{-1}$) are among the highest reported for the Laurentian Great Lakes. Previous denitrification measurements in offshore zones of the Great Lakes vary over four orders of magnitude, with western and central Lake Erie exhibiting the highest rates at 51 ± 41 µmol N m$^{-2}$ h$^{-1}$ (Small et al., 2014; 2016). Nearshore zones, bays, and river mouths have been observed as areas of enhanced denitrification compared to offshore zones (McCarthy et al., 2007; Small et al., 2014; 2016), reinforcing that Sandusky Bay and other shallow coastal areas have the potential to act as hotspots of N removal in the Great Lakes system.

Anammox activity in Sandusky Bay was highly variable, even among replicate sediment cores from the same site and date. The range of observed anammox rates (0-30.75 µmol N m$^{-2}$ h$^{-1}$) is similar to those reported for shallow estuarine environments (Dong et al., 2009; 2011; Hsu and Kao, 2013). Marked variability may be characteristic of anammox activity in freshwater environments, even across small spatial and temporal scales (Yoshinaga et al., 2011; Zhu et al., 2013; 2015). Anammox made up an average of 14 % of total N removal across the sampling period, indicating anammox activity may be





typical of shallow estuarine and freshwater systems (Thamdrup and Dalsgaard, 2002; Dalsgaard et al., 2005; Schubert et al., 2006; McCarthy et al., 2016). A recent study of 16s RNA showed that anammox taxa were present in sediment across western and central basins of Lake Erie (Small et al., 2016). Collectively, the anammox rate and genomic data demonstrate that anammox has the potential to be an appreciable N removal pathway in this and other nearshore regions within the Great

Lakes.

Rates of $N_2O$ production in Sandusky Bay were consistently low relative to rates of denitrification and anammox and comprised an average of 2 % of N removal. The highest rates of $N_2O$ production occurred when $NO_3^-$ concentrations were highest, suggesting that nutrient delivery from the Sandusky River drives benthic $N_2O$ production and release in Sandusky

Bay. Moreover, active N cycling in this and other Great Lakes estuaries (McCarthy et al., 2007; Salk et al., 2016; Small et al., 2016) suggests that atmospheric emissions from these locations may make up the majority of $N_2O$ emissions from the Great Lakes.

Whereas denitrification and anammox were the primary drivers of dissimilatory N uptake in Sandusky Bay, hydraulic

residence time had a marked effect on N removal. In 2015, estimates of water residence time when discharge peaked in June and early July was as low as eight days. By late July, however, hydraulic residence time increased to several months and continued to increase as discharge remained low for the remainder of the summer and early fall. Although N removal rates were greatest when $NO_3^-$ concentrations were highest, the capacity for N removal to substantially deplete $NO_3^-$ was hindered by the short hydraulic residence time within the Bay. For instance, high N removal rates in late June 2015 coincided with

high $NO_3^-$ concentrations, but hydraulic residence times of 1-2 weeks would enable only a small fraction of $NO_3^-$ to be permanently removed from the system prior to release into Lake Erie. The depletion of N in Sandusky Bay occurred when water residence time lengthened to several months, which provided the opportunity for assimilatory and dissimilatory processes to extensively consume $NO_3^-$. Although phytoplankton uptake represents another significant N consumption process, the sharp decline in dissolved N:P ratios during this period (Fig. 2g, 2h) indicates that dissimilatory processes (i.e.,

denitrification, anammox, $N_2O$ production) were important drivers of $NO_3^-$ concentration decline. Thus, Sandusky Bay acts as a conduit for N delivery from the Sandusky River to Lake Erie when hydraulic residence time is short but acts as a filter for $NO_3^-$ during periods of long hydraulic residence time. Export of nutrients from Sandusky Bay to Lake Erie during periods of high discharge is illustrated by concomitant $NO_3^-$ concentration spikes in Sandusky Bay and at the Lake Erie Bells station in July 2015 (Fig. 2a). Conversely, a signal of $NO_3^-$ export from Sandusky Bay into Lake Erie is lost during periods of low

discharge, with the Bells station displaying higher $NO_3^-$ concentrations than bay stations in 2016 (Fig. 2b). The capacity for Sandusky Bay to act as an alternating conduit and filter for nutrients is consistent with observations in other river mouths and estuaries in the Great Lakes (McCarthy et al., 2007; Larson et al., 2013; Conroy et al., 2017).



### 4.3 Phytoplankton N Acquisition

*Planktothrix*-dominated phytoplankton blooms (Davis et al., 2015) were evident in Sandusky Bay in both 2015 and 2016. Chl *a*, a proxy for total phytoplankton biomass (Becker et al., 2009; Millie et al., 2009; Davis et al., 2012), reached maximum levels approximately one month after maximum $NO_3^-$ and $PO_4^{3-}$ concentrations were observed (Fig. 2). This offset

in peak nutrient availability and peak phytoplankton biomass has been observed in other years as well (Conroy et al., 2017). The coincidence of peaks in chl *a* with low river discharge is consistent with the idea that long hydraulic residence times create a stable physical environment in which primary producers can flourish (Michalak et al., 2013). Dissolved N:P ratios during the period of highest phytoplankton abundance approached or exceeded the threshold for N limitation, suggesting that *Planktothrix* is successful in acquiring N during periods of scarcity.

Given that *Planktothrix* is an effective competitor for DIN (Conroy et al., 2007), $NO_3^-$ and $NH_4^+$ were investigated as sources of N for HABs in Sandusky Bay. Indeed, $NO_3^-$ and $NH_4^+$ uptake were active throughout the summer, demonstrating that low concentrations do not equate to the absence of an actively cycling DIN pool. Transient pools of $NH_4^+$ generated via water column recycling (Chaffin and Bridgeman, 2014; Donald et al., 2011; 2013; Davis et al., 2015) or sediment regeneration

(Paerl et al., 2011; McCarthy et al., 2016) could help to support persistence of *Planktothrix* blooms in the late summer. Uptake rates were proportional to the respective concentrations of substrates, suggesting that although $NH_4^+$ is a less energetically costly source of N, $NO_3^-$ is utilized preferentially by phytoplankton in this system owing to a greater relative abundance.

Remineralization was not specifically measured in $NH_4^+$ uptake assays, but it is likely that $NH_4^+$ is rapidly recycled in this system. $NH_4^+$ remineralization has the potential to violate the assumptions of the $NH_4^+$ uptake assay, as it would cause dilution of the $^{15}N$-enriched $NH_4^+$ pool. This possibility was accounted for by computing a hypothetical dilution due to remineralization and by assuming a remineralization rate equivalent to the $NH_4^+$ uptake rate. As $NH_4^+$ concentrations are relatively constant, we expect that equating rates of remineralization and uptake is a reasonable expectation. In our estimates,

we assume an immediate dilution rather than a progressive dilution throughout the duration of the incubation and an isotopic composition of remineralized $NH_4^+$ equal to that of particulate N. Thus, our calculation represents the maximum possible dilution that could take place during $NH_4^+$ uptake incubations. The resulting isotope dilution would result in an overestimation of $NH_4^+$ uptake by $41\pm11$ % (mean ± SD). While this is an important component to consider for a system with rapidly cycling N, it is worth pointing out that even assuming maximum remineralization rates, $NH_4^+$ uptake is

exceeded by $NO_3^-$ uptake and N fixation in this system.

N fixation was a major N uptake process in Sandusky Bay, often exceeding rates of $NO_3^-$ and $NH_4^+$ uptake (Fig. 3). Previous work has shown that the phytoplankton community responsible for water column N fixation in Sandusky Bay is comprised




largely of *Aphanizomenon* and *Dolichospermum* (Davis et al., 2015). Areal N fixation rates (309.5-906.9 µmol N m$^{-2}$ h$^{-1}$ in 2015 and 0.2-187.8 µmol N m$^{-2}$ h$^{-1}$ in 2016) are within the range of those observed in eutrophic lakes (9.2-421.2 µmol N m$^{-2}$ h$^{-1}$; Howarth et al., 1988 and references therein), with the exception of extremely high N fixation rates measured in May, July, and August 2015. The occurrence of high N fixation rates under N limitation in eutrophic Sandusky Bay is not surprising, but the observation of N fixation rates exceeding NO$_3^-$ and NH$_4^+$ uptake rates on occasions when both substrates were readily available is unanticipated.

On several occasions during periods of low discharge, riverine DIN loading alone could not meet phytoplankton uptake demands (Table 1). Consequently, fixation and subsequent recycling of N represent an additional potential source of bioavailable N for cHABs in Sandusky Bay. For dates when comparisons are available, N fixation comprised 23.7-85.4 % of total phytoplankton N uptake (NO$_3^-$ uptake + NH$_4^+$ uptake + N fixation) in the inner and outer bay. This range is typical of eutrophic lakes, in which N fixation has been observed to comprise 5.5-82.0 % of N inputs (Howarth et al. 1988 and references therein). Given that a high proportion of N in the system is supplied via N fixation, even small transfers of N from diazotrophs could represent an important N source for *Planktothrix*. Specifically, DIN leaking out of actively N-fixing cells or from decomposing cells has the potential to supplement N supply for non-diazotrophs (Ohlendieck et al., 2000; Beversdorf et al., 2013). Thus, both riverine N loading and N fixation represent sources of bioavailable N that *Planktothrix* may scavenge during periods of N limitation in Sandusky Bay.

Rates of N fixation were not correlated with DIN concentrations, a surprising outcome considering that DIN uptake is a less energetically costly process and is predicted to outcompete N fixation when DIN is available (Holl and Montoya, 2005). While this result is unexpected, previous work has shown that the presence of DIN suppresses the synthesis of the nitrogenase complex but not the activity of the existing enzyme (Fogg, 1971; Wolk, 1973; Chang et al., 1980), and high concentrations of NO$_3^-$ only partially suppress heterocyst formation (Ogawa and Carr, 1969; Ohmori and Hattori, 1972). Indeed, N fixation in the presence of DIN and at dissolved N:P ratios greater than 16 have been observed elsewhere (Chen et al., 1996; Spröber et al., 2003; Voss et al., 2004; Moisander et al., 2008; Gao et al., 2014). Given the dynamic nature of hydrology and riverine DIN loading in this and other estuarine systems, there may be energetic gains from maintaining N fixation machinery that can be utilized quickly following sudden swings in N availability (Moisander et al., 2012). While the precise causes of high rates of N fixation in this hypereutrophic system are not readily apparent, the occurrence of this process indicates that Sandusky Bay not only acts as a conduit for riverine DIN loading to Lake Erie but also a source. Therefore, effective management of this system to minimize N export must not only evaluate N loading and loss processes but also the introduction of N via N fixation.



## 4.4 N Budget and the N Pendulum

Sandusky Bay displays dynamic swings in hydraulic and nutrient regimes yet consistently develops seasonal N limitation and blooms of *Planktothrix* in the late summer. An examination of the N budget in this system will help to characterize the role of this Great Lakes estuary in mediating N delivery to Lake Erie (Conroy et al., 2017). On the whole, Sandusky Bay is a

source of N to Lake Erie, with a small to large surplus of N delivered downstream that is highly dependent on Sandusky River discharge (Fig. 2, Table 1). During portions of the year when riverine DIN loading is low, N fixation supplements DIN loading to meet assimilatory and dissimilatory N demands. N fixation thus represents a large and crucial balance for the N budget in Sandusky Bay. Phytoplankton uptake is the predominant N uptake, whereas sediment N removal processes represent a modest N sink relative to the high input of $NO_3^-$ from the Sandusky River. However, the fates of N for

assimilatory and dissimilatory processes are distinct, which could have implications for N limitation and HABs in Sandusky Bay. Assimilatory processes represent temporary N sinks that retain N in the system to be recycled or delivered downstream. The dominance of assimilatory processes suggests that although DIN concentrations are often low in Sandusky Bay, there is an actively cycling N stock within the phytoplankton community that may be utilized by *Planktothrix*. Dissimilatory sinks, although on a smaller magnitude, represent a permanent N sink that may have a greater influence on the development of N

limitation than assimilatory sinks.

An important consideration for the N budget in Sandusky Bay is that N supply and N depletion are temporally separated. Sandusky Bay undergoes rapid and dramatic seasonal transitions from excess N abundance to N limitation that are consistent from year to year (Conroy et al., 2007; Davis et al., 2015; Conroy et al., 2017). During periods of high discharge and N

loading, sources of N outweigh sinks, resulting in pulses of DIN delivery into Lake Erie. When N sources are small under long hydraulic residence time, assimilatory and dissimilatory processes deplete DIN, and relatively little DIN is flushed into Lake Erie. In this manner, Sandusky Bay oscillates between acting as a strong N source and a weak N source, creating a N pendulum. This N pendulum may be a common feature of other estuarine systems, most notably Narragansett Bay (Fulweiler et al., 2007; Fulweiler and Heiss, 2014). The pendulum action of N cycling in Sandusky Bay indicates that the system may

be oscillating between alternate states, and the capacity for the bay to act as a net N sink has been overwhelmed by riverine N loading.

Future projections suggest that climate change will create conditions that are likely to intensify the swings of the N pendulum in Sandusky Bay and other coastal systems as well. Overall precipitation in the watershed is expected to increase

but will be more episodic in nature and dominated by incidences of extreme precipitation (Prein et al., 2017). Moreover, increases in precipitation are predicted to be accompanied of enhanced riverine N loading in the near future (Sinha et al., 2017). These changing hydrologic patterns have substantial implications for N cycling and cHABs in Lake Erie. Periods of low discharge will likely stimulate N loss processes, N limitation, and the competitive advantage of *Planktothrix* in

Sandusky Bay, while episodic high discharge events will favor export of nutrients that may support cHABs in the central basin of Lake Erie and exacerbate water quality issues downstream, including hypoxia in the central basin of Lake Erie and the St. Lawrence Estuary (Lehmann et al., 2009; Michalak et al., 2013). As a result of climate change, the N pendulum in Sandusky Bay is thus likely to swing in favor of increased total annual N export. Similar climate change-driven shifts in water quality are anticipated in other coastal systems such as the Gulf of Mexico, where enhanced riverine N delivery and predicted increases in magnitude and timing of precipitation will make hypoxia mitigation efforts even more difficult (Turner et al., 2008; 2012). Sandusky Bay may thus serve as a harbinger for what can be expected in many coastal systems that are responding to climate change and increases in N loading.

## Data Availability

Data and associated content for this manuscript can be found at https://github.com/KateriSalk/SanduskyBay_NitrogenCycle.git.

## Author Contribution

This study was designed by KS, NO, GB, and RMM. Field sampling was carried out by KS, GB, RMM, JC, and associated research groups. Sample and data analysis were carried out by KS, NO, and JC. KS prepared the manuscript with contributions from NO, GB, RMM, and JC.

## Acknowledgements

We thank Taylor Tuttle, Emily Davenport, Hasand Gandhi, Kristen Slodysko, Erica Fox, Kat Rossos, Callie Nauman for their assistance in the field and laboratory. Thank you to Silvia Newell and Mark McCarthy for assistance with MIMS analysis, and Peggy Ostrom, Stephen Hamilton, and Silvia newell for feedback on the manuscript. This research was supported by the NSF Graduate Research Fellowship [No. 497 DGE1424871], the Michigan State University (MSU) College of Natural Science Hensley Fellowship, the MSU Rose Fellowship in Water Research, the MSU WaterCube program, the Ohio Department of Higher Education's Harmful Algal Bloom Research Initiative [No. R/HAB-2-BOR], and the Ohio Sea Grant College Program [No. R/ER-110].

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





**Tables**

Table 1. Total range in N sources, assimilatory N uptake processes, and dissimilatory N sinks in Sandusky Bay.

| Process | Minimum (kg d$^{-1}$) | Maximum (kg d$^{-1}$) |
|---|---|---|
| **Sources** | | |
| DIN loading | 1 | 355,054 |
| N fixation | 2,143 | 102,179 |
| *Total* | *2,144* | *457,233* |
| | | |
| **Assimilatory N uptake** | | |
| $NH_4^+$ uptake | 267 | 6,264 |
| $NO_3^-$ uptake | 1,360 | 104,709 |
| *Total* | *1,627* | *110,973* |
| | | |
| **Dissimilatory N uptake** | | |
| Denitrification | 274 | 1,769 |
| Anammox | 14 | 221 |
| $N_2O$ production | 2 | 64 |
| *Total* | *290* | *2,054* |



**Figures**

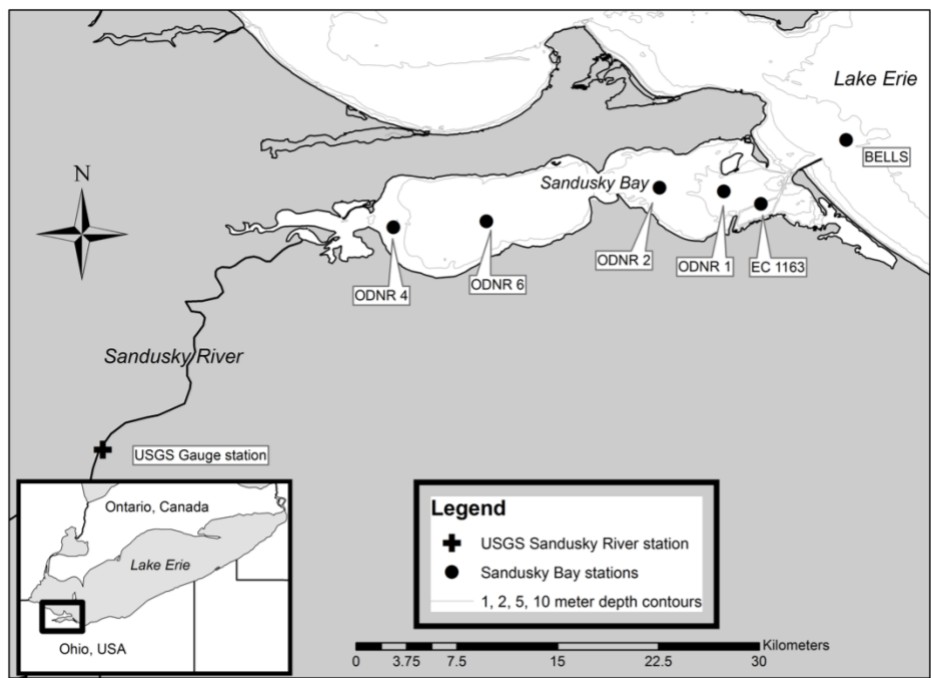

**Fig. 1: Sampling locations in Sandusky Bay (circles) and Sandusky River monitoring station (USGS monitoring station 04198000, cross).**





**Fig. 2: (a-b) NO₃⁻, (c-d) NH₄⁺, (e-f) PO₄³⁻, (g-h) N:P (NO₃⁻ + NH₄⁺ : PO₄³⁻) ratio, (i-j) chlorophyll *a*, and (k-l) Sandusky River discharge, in 2015 and 2016, respectively. Six Sandusky Bay sites are presented: inner bay (ODNR and ODNR6, black filled symbols), the outer bay (ODNR2, ODNR1, and 1163, gray filled symbols), and a site outside the bay in the central basin of Lake Erie (Bells, gray open symbols). NO₃⁻ concentrations in the Sandusky River are also presented as a dotted line (a-b). Note the consistent scales between years for single variables (exception: discharge) but differing scales among variables. The dotted line in (g-h) indicates a N:P ratio of 16 (note log scale on y axis).**





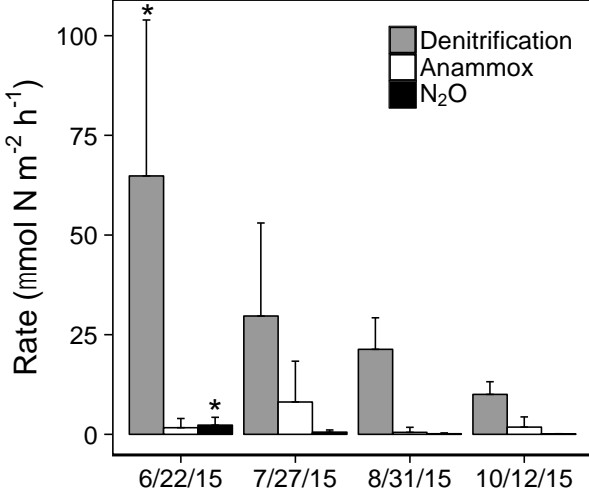

**Fig. 3: Denitrification, anammox, and N₂O production rates at station 1163. Error bars represent +1 SD. Asterisks (*) indicate statistical significance for a single process among dates at $p < 0.05$, as indicated by a one-way ANOVA and a subsequent Tukey's HSD post-hoc test.**



**Fig. 4: (a) NO₃⁻ uptake rates, (b) NH₄⁺ uptake rates, and (c) N fixation rates in Sandusky Bay. Rates were measured at stations ODNR4 (black), 1163 (gray), and Bells (white). Note the differing y-axis scales among panels. Error bars represent +1 SD. Letters indicate statistically significant groupings at *p* < 0.05, as indicated by a two-way interaction effects ANOVA and a subsequent Tukey's HSD post-hoc test. Carrot symbols (^) indicate incubations in which the ¹⁵N addition exceeded 10 % of the ambient NO₃⁻ or NH₄⁺ concentration, owing to unexpectedly low ambient concentrations.**





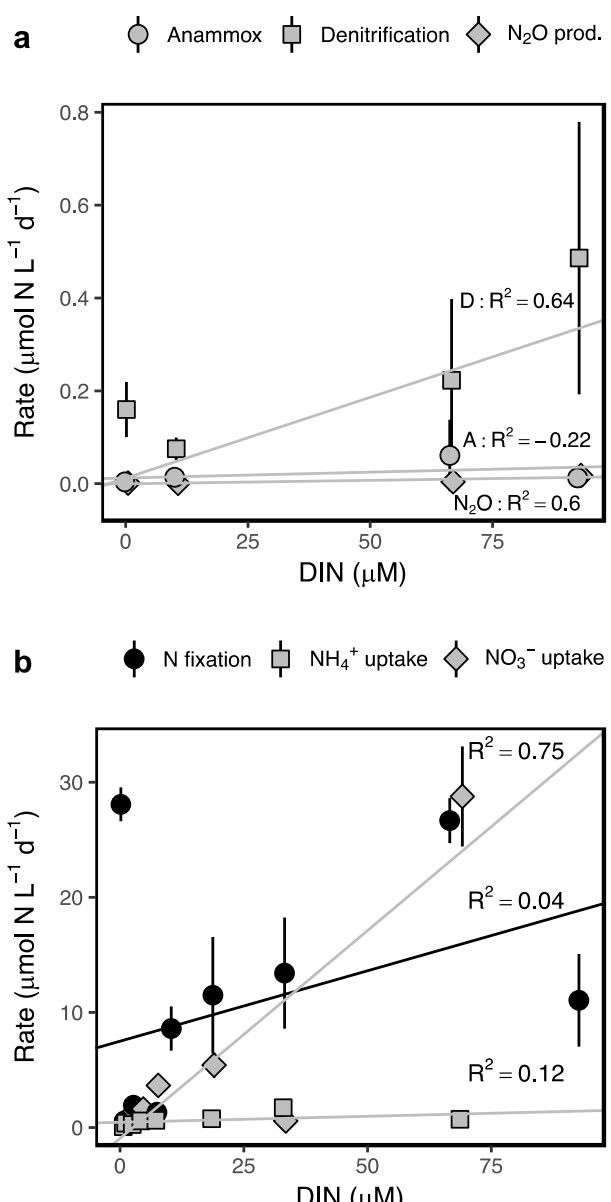

Fig. 5: Rates of (a) dissimilatory and (b) assimilatory N cycling processes as a function of DIN concentration ($NH_4^+ + NO_3^- + NO_2^-$). Note the differing y-axis scales among panels. Error bars represent ± 1 SD. $R^2$ represents the adjusted $R^2$ for each linear regression.

