# Peer review of "Nitrogen cycling in Sandusky Bay, Lake Erie: Oscillations between strong and weak export and implications for harmful algal blooms"

_Biogeosciences, 2017_

## Referee Comment (RC1) · Anonymous Referee #1 · 26 Jan 2018

I should start by noting that I am not familiar with Lake Eyrie, and rather review this as someone who works on largely coastal nutrient cycling. From this perspective I found this paper interesting and was impressed by the wide range of techniques brought to bear on the issue of nutrient and particularly nitrogen cycling in this system. Possibly because I am not familiar with the regional water quality management issues, I have one rather general comment on the paper and particularly its stated aims. This paper presents an impressive study of the factors controlling nutrient cycling in Sandusky Bay and how this relates to blooms as measured by chlorophyll. Throughout the manuscript the authors refer to cHABs and to Planktothrix, although all of the measurements made are of chlorophyll abundance. My rather incomplete understanding of HABs is that

they are an episodic phenomena which may be related to nutrient loadings, but also to a variety of other factors, and furthermore even blooms of the same species may or may not produce toxins and associated harmful products (e.g. Berdalet et al 2017 Oceaongraphy 30, 46-57.) I would therefore suggest the authors be cautious about the way they associate chlorophyll blooms and HABs. I note Davies et al 2015 cited here does say that Planktothrix dominates completely here, and if that is the case this can be stated here to address my point, but HABs are such a high profile policy issue that I think the terminology needs to be used with care. That aside, I find this paper of considerable interest. The scale of interannual variation in flow and nutrient loadings is enormous and provides an interesting setting in which to evaluate biogeochemical responses, so I do not see that the paper has to attach itself so closely to the HAB issue to be of general interest. Specific points Throughout the manuscript the authors describe the bay as "Great Lakes estuary" – this may be the way the term is used in this region, but my understanding is that an estuary involves mixing of fresh and salt water. The denitrification results are impressive and interesting and I rather like the way the authors compare them across a wide range of systems – this is a process that presumably should be microbially similar in fresh and salt waters. I would note on p11 that this process does of course also depend on carbon supply (as well as nitrate) and it has been argued that the relative significance of the various N removal process depend on the C/nitrate ratios. P12 I think it is widely accepted that N2O production is a bi-product of denitrification (and nitrification) and so it is inevitable that N2O production will be much lower than denitrification rates, and of course anammox does not produce N2O. It also follows that N2O production will increase with denitrification and if the latter is linked to nitrate supply, then N2O production will increase with nitrate. Hence all of this (lines 6-12) is rather obvious, but then I would question the logic and link to the last sentence in paragraph 2 p12 "Moreover, active N cycling. . ." which is not justified. Section 4.3 The authors make the very important point that they did not measure remineralisation. I accept completely that there is only so much they can do, so I do not criticise that omission. The issue of remineralisation is discussed in terms

of its implications for ammonium uptake which is fine. However, the other important point is that in terms of impacts on the N budget, the phytoplankton uptake measured is a gross rate and really it is the net rate that controls the budget and this needs to be discussed further in section 4.3 and 4.4. If the phytoplankton N taken up is regenerated within the bay, it is not necessarily a sink at all. The other intriguing feature of this section is the scale of nitrogen fixation in a nitrogen rich system. As the authors point out there has long been a conceptual assumption that nitrogen fixation takes place only in response to nitrogen limitation, and indeed in the marine community that nitrogen stress limiting N fixation is assumed to occur at concentrations of a few micromolar, way below the ambient concentrations in this system. The results here and elsewhere should encourage a thorough investigation of nitrogen fixation and its controls. I would note that it also needs a supply of P and Fe, which may also exert a control in this system. Section 4.4 I note the interesting argument that this system is switching from a source to a sink and this is really controlled by hydraulic residence time, and I agree with that. The outer boundary of the bay is really artificial anyway and nitrogen removal by denitrification will continue into the main lake, regardless of the situation within the bay. I also understand the point that climate change may make the flow variations more extreme at this location. However, I am not sure it is straightforward to extrapolate this argument to other "coastal areas" (line 29). In these tides play a major role in moving water around. In addition my understanding of both Narragansett Bay (line 23, where I think the flip in behaviour has been attributed to changes in carbon supply) and The Gulf of Mexico (line 5 p16, which is hypoxic driven by Mississippi nutrient inputs) is that there is no reason to expect the kind of water flow driven "pendulum" there.

---

## Referee Comment (RC2) · Anonymous Referee #2 · 13 Feb 2018

The manuscript reports on measurements of nitrogen cycling and assimilation processes in Sandusky Bay. The work important because it reports a simultaneously measured rates of denitrification, nitrogen fixation and loads to the bay which allows the relative importance of the processes to be assessed. It was found that in general nitrogen fixation rates were higher than nitrogen loss through denitrification. As such, the bay generally acts as a net source of nitrogen to Lake Eyrie rather than a sink. My only major comment is that I think that the budget could have been presented more clearly to better highlight how the bay modulates nitrogen inputs to Lake Eyrie.

Specific comments

Abstract I suggest referring to dissimilatory nitrate reductive processes as opposed to assimilation

Use of 'pendulums' not really the right term in my opinion. As elaborated on below, I think the key point is that the system is a modulator of nutrient inputs.

Estuary, in a freshwater lake? How about mixing zone?

Pg 3 line 5 – I don't think estuary is generally accepted as a term for rivers entering freshwater lakes – I suggest mixing zone.

Methods Isotope analysis delta15N values are mentioned in the methods, why? The isotope pairing equations use excess ratios of M/Z 29/28 and 30/28 for N2 and 45/44 and 46/46 for N2. I suggest deleting all ref to del 15N and explaining which masses were monitored and how excess ratios were calculated. It also not clear why N2 was also measured with MIMS or how these data were used.

Phytoplankton N uptake 15NH4/NO3 contamination of 15N2. You state that uptake of contamination would have made up less than 5% of measured rates. This depends on the rates. Is this even the case for the lowest measured rates? The main thing that convinced me your data were probably ok, was the fact you could measure low rates in 2016.

Budget I don't think converting sediment process rates to volumetric rates is meaningful – these should either be shown as areal rates or total mass for the whole system.

Line 5 pg 8 – I agree with your point about TKN in the river, but what about in Sandusky Bay? As mentioned below, I think the system is really a modulator that converts NOx to organic matter and this will be shown clearly in the TKN data if available.

For the nitrate loads, at what time interval were concentration and flow measured? How were these data interpolated to calculate loads?

Discussion N removal processes. This was generally good and I agree with the arguments. I felt, however this section could have been a little more quantitative. For example, it is argued that the increase in the N:P was driven in a large part by denitrification. I suggest the authors undertake a back of the envelope calculation to show how the change in the mass of $NO_3$ in the water column over this period of $NO_3^-$ drawdown compares with total denitrification measured over the same period. Also of relevance here is that Dw (water column driven denitrification) and Dn (water column driven denitrification) are not reported. The breakdown of these is important when considering the drawdown rate of NOx.

Budget I think there was a missed opportunity with the budget to integrate the findings a little more clearly. I suggest that for each period process rates were measured, a budget be undertaken (could be daily or perhaps monthly basis). These budget terms could then graphed to highlight the change from high catchment inputs to high internal inputs via N fixation as flows decreased through to August. This would also highlight the relatively minor importance of denitrification as a sink compared to the inputs. Although the phytoplankton assimilation measurements are a nice part of the paper, I don't think they can be used meaningfully in the budget because they were taken in 2016 when phytoplankton biomass was higher.

I think the discussion at line 20 on pg 15 could also talk a little more about the system as a transformer of nitrogen importing DIN and exporting algal biomass as well as N derived from nitrogen fixation. At the moment it is a bit repetitive and not as interesting as it could be. I don't really think the term N pendulum is correct, it really modulates the inputs depending on residence time, with a net export of nitrogen from nitrogen fixation. This finding is consistent with a previous study of a shallow eutrophic lake which often showed net exports of total nitrogen, most likely due to nitrogen fixation.

Cook, P.L.M., K.T. Aldridge, S. Lamontagne, and J.D. Brookes. (2010). Retention of nitrogen, phosphorus and silicon in a large semi-arid riverine lake system. Biogeochemistry, 99: 49-63.

The last paragraph of the discussion is quite speculative, I suggest remove.

Figures Figure 2, micro symbol now appears as milli.

Figure 3 micro symbol as above

Figure 4 the letters showing statistically significant groupings are unclear. ˆ is carat, not carrot

Figure 5a. Why are these rates reported volumetrically? They should be areal as for Fig 3.

---

## Author Comment (AC1) · 22 Mar 2018

Reviewer comment: Issues in associating chlorophyll concentrations with HABs Author response: This point is well taken. Originally, the authors had planned to publish an accompanying manuscript on the genomic composition of the community in Sandusky Bay, which more clearly illustrates that the connection between chlorophyll and HABs in the system is justified. However, the authors are now planning to add a brief section to the manuscript on the metagenomic analysis of the phytoplankton community, which demonstrates a dominance by Planktothrix and the presence of N-fixing cyanobacteria during periods of high rates of N fixation. This will include a paragraph each in the

methods, results, and discussion, with an accompanying figure of the species composition in Sandusky Bay.

Reviewer comment: Use of the term "Great Lakes estuary" or "freshwater estuary" may not be appropriate Author response: "Freshwater estuary" is indeed a class of estuary, as defined by the U.S. National Oceanic and Atmospheric Administration (https://oceanservice.noaa.gov/education/kits/estuaries/media/supp_estuar05e_fresh.html). These systems are common in the Laurentian Great Lakes, and they are characterized by gradients and hydrology similar to marine estuaries (including conductivity gradients). Another system in Lake Erie, Old Woman Creek, is a freshwater estuary in the National Estuarine Research Reserve System. Although this term is unusual, we prefer to use it to illustrate common processes among these systems and marine estuaries (i.e., freshwater estuaries often behave more like marine estuaries than lakes).

Reviewer comment: p. 11 – denitrification relies on carbon supply as well as nitrate, potential dependence on C:nitrate ratios Author response: Given the high rates of primary productivity in this system and the high carbon content of the sediments (e.g., Ostrom et al. 2005), the consideration of C limitation on denitrification was not a major concern for this study. Additionally, a recent study showed that sediment C:N ratios were not significant predictors of denitrification rates across the Great Lakes and in Lake Erie (Small et al. 2016). Considerations of C supply will be mentioned in the revised manuscript in the methods section where Sandusky Bay is introduced as the study system. Ostrom et al. 2005. Evaluation of primary production in Lake Erie by multiple proxies. Oecologia 144: 115-124. doi: 10.1007/s00442-005-0032-5 Small et al. 2016. Large differences in potential denitrification and microbial communities across the Laurentian great lakes. Biogeochemistry 128: 353. doi: 10.1007/s10533-016-0212-x

Reviewer comment: p. 12 – obvious comments in lines 6-12 on N2O production as byproduct of denitrification, but logic linking this to last sentence in paragraph 2 is not

justified Author response: These lines will be incorporated into the first paragraph of section 4.2, which will help to improve the flow and more clearly tie the results to the final sentence emphasizing that shallow regions may make up the majority of N2O emissions from the Great Lakes.

Reviewer comment: Section 4.3 – Impact of remineralization on N budget needs to be discussed further, particularly the distinction between the gross and net rate of phytoplankton uptake (is uptake a sink?) Author response: Remineralization is discussed on p. 13 lines 20-30, and the issue of phytoplankton assimilation as only a temporary sink is discussed on p. 15 lines 9-15. The reviewer points out an opportunity to make this connection more explicit. As a result, we will revise those paragraphs to more thoroughly describe the distinction between gross and net rates of phytoplankton N assimilation, and the capacity for recycling to retain N in the system.

Reviewer comment: Section 4.3 – These results encourage a thorough investigation of the controls of nitrogen fixation, including a supply of P and Fe Author response: The reviewer points out an interesting and relevant consideration, which the authors considered as well in the preparation of this manuscript. While Fe limitation of N fixation has precedent in the literature in several systems, we dismissed this mechanism for two reasons: (1) as the Sandusky River discharges into the bay, it brings a high sediment load that is expected to have a sufficient trace metal load to support phytoplankton demands, and (2) the sediments in Sandusky Bay are anoxic, and it is likely that nighttime sediment oxygen demand is high enough that transient anoxia develops at the sediment-water interface, thus enabling P and reduced Fe to be released into the water column and be taken up by phytoplankton. These internal loading events, while outside the scope of this study, have been captured on several occasions over the course of multi-year surveys of the bay. While P limitation was certainly at play during portions of the season in both years, the substantial phytoplankton biomass and high N fixation rates indicated that P was likely not a driving factor for the activity of N fixation. We will include an additional section on cyanobacterial community metatranscriptomics in the revised manuscript (see author response to comment #1), which lends additional support to the biogeochemical measurements of N fixation.

Reviewer comment: Section 4.4, line 29 – not sure it is straightforward to extrapolate this argument to other coastal areas (likely that water flow is not operating as a pendulum in other coastal systems, where tides and other factors might be stronger controls) Author response: This is a great point that other factors come into play beyond changes to precipitation and N loading, which in turn make it complicated to extrapolate results from Sandusky Bay to other systems. Despite the potential differences, many estuarine systems experience strong river input and therefore may experience similar hydrology-driven swings in nutrient availability and biogeochemistry (thus the pendulum may apply in other systems as well). The authors will edit the language in this paragraph to reflect this.

---

## Author Comment (AC2) · 22 Mar 2018

Reviewer comment: Abstract I suggest referring to dissimilatory nitrate reductive processes as opposed to assimilation Author response: Assimilatory (phytoplankton uptake) and dissimilatory (microbial N reduction) processes are considered separately, and this language will be revised to more clearly reflect this distinction.

Reviewer comment: Use of 'pendulums' not really the right term in my opinion. As elaborated on below, I think the key point is that the system is a modulator of nutrient inputs. Author response: Based on our observations of the "swings" in hydrology, nutrient inputs, and capacity for the bay to transform N, we chose the term "pendulum" as

a descriptor for how the system functions. Given that this is a strong visual metaphor, we would prefer to retain this term in the manuscript. We are willing to discuss with the editor the possibility of changing this term (e.g., modulator, oscillations) if desired.

Reviewer comment: Pg 3 line 5 – I don't think estuary is generally accepted as a term for rivers entering freshwater lakes – I suggest mixing zone. Author response: "Freshwater estuary" is indeed a class of estuary, as defined by the U.S. National Oceanic and Atmospheric Administration (https://oceanservice.noaa.gov/education/kits/estuaries/media/supp_estuar05e_fresh.html). These systems are common in the Laurentian Great Lakes, and they are characterized by gradients and hydrology similar to marine estuaries (including conductivity gradients). Another system in Lake Erie, Old Woman Creek, is a freshwater estuary in the National Estuarine Research Reserve System. Although this term is unusual, we prefer to use it to illustrate common processes among these systems and marine estuaries (i.e., freshwater estuaries often behave more like marine estuaries than like lakes).

Reviewer comment: Methods Isotope analysis delta15N values are mentioned in the methods, why? The isotope pairing equations use excess ratios of M/Z 29/28 and 30/28 for N2 and 45/44 and 46/46 for N2. I suggest deleting all ref to del 15N and explaining which masses were monitored and how excess ratios were calculated. It also not clear why N2 was also measured with MIMS or how these data were used. Author response: Methods for the determination of isotopic composition and the calculations associated with the IPT will be revised for clarity. Equation 1 will be removed (as the reviewer points out, we are working directly with ratios, not deltas). Details on calculating mass ratios will be added. A description of why N2 was measured with MIMS will be added (concentration measurements were not possible on IRMS given our approach).

Reviewer comment: Phytoplankton N uptake 15NH4/NO3 contamination of 15N2. You state that uptake of contamination would have made up less than 5% of measured rates. This depends on the rates. Is this even the case for the lowest measured rates?

The main thing that convinced me your data were probably ok, was the fact you could measure low rates in 2016. Author response: The 5% value for potential contamination is the maximum potential contamination across all measurements, from low to high rates. We calculated the possible overestimation of N fixation given the level of potential contamination reported in Dabundo et al. (2014), and 5 % was the highest calculated value across the dataset. This statement will be revised to clarify this point.

Reviewer comment: Budget I don't think converting sediment process rates to volumetric rates is meaningful – these should either be shown as areal rates or total mass for the whole system. Author response: Volumetric rates will be converted to areal rates for the budget calculations.

Reviewer comment: Line 5 pg 8 – I agree with your point about TKN in the river, but what about in Sandusky Bay? As mentioned below, I think the system is really a modulator that converts NOx to organic matter and this will be shown clearly in the TKN data if available. Author response: Supplemental data collected by the Heidelberg NCWQR and coauthor J. Chaffin indicates that indeed, TKN concentrations at sites in Sandusky Bay are generally higher than at the Sandusky River monitoring station. This supports the point that assimilatory processes (and associated recycling) convert DIN to organic N in the bay, thus modulating the delivery of N downstream. This is consistent with our observations that assimilatory processes are the dominant N uptake process, recycling N within the Bay and (partially) converting downstream N delivery from inorganic to organic forms. We will cite this observation as unpublished data and consider adding a supplemental figure to illustrate the TKN data for the river and bay sites.

Reviewer comment: For the nitrate loads, at what time interval were concentration and flow measured? How were these data interpolated to calculate loads? Author response: Daily nitrate loads were calculated based on daily discharge and nitrate concentration data from the Heidelberg NCWQR. Direct comparisons were made for days when assimilatory and dissimilatory N transformations were measured.

Reviewer comment: Discussion N removal processes. This was generally good and I agree with the arguments. I felt, however this section could have been a little more quantitative. For example, it is argued that the increase in the N:P was driven in a large part by denitrification. I suggest the authors undertake a back of the envelope calculation to show how the change in the mass of NO3 in the water column over this period of NO3- drawdown compares with total denitrification measured over the same period. Author response: The capacity of denitrification and anammox to draw down NO3 concentrations is qualitatively described on p. 12 lines 19-23. Calculated values for the "small fraction," as descried in these lines will be added to the revised manuscript to make this description more quantitative.

Reviewer comment: Discussion N removal processes. Also of relevance here is that Dw (water column driven denitrification) and Dn (water column driven denitrification) are not reported. The breakdown of these is important when considering the draw-down rate of NOx. Author response: Given the shallow and well-mixed nature of this system (mean depth 1.6-2.6 m), water column denitrification was not considered as an appreciable denitrification source. This assumption was borne out in the IPT calculations, which enable distinction between Dw and Dn and confirmed that Dw was not active. These detailed results were not included in the manuscript, but a statement describing the assumption and confirmation of negligible water column denitrification will be added.

Reviewer comment: Budget I think there was a missed opportunity with the budget to integrate the findings a little more clearly. I suggest that for each period process rates were measured, a budget be undertaken (could be daily or perhaps monthly basis). These budget terms could then graphed to highlight the change from high catchment inputs to high internal inputs via N fixation as flows decreased through to August. This would also highlight the relatively minor importance of denitrification as a sink compared to the inputs. Although the phytoplankton assimilation measurements are a nice part of the paper, I don't think they can be used meaningfully in the budget because

they were taken in 2016 when phytoplankton biomass was higher. Author response: The reviewer brings up a good point that the budget calculations could be more effectively integrated to illustrate the system more clearly. Table 1 will be expanded to include daily budgets (i.e., proportion of DIN loading consumed by each measured process), and a graph will be added to illustrate the shifts in sources and sinks temporally. The authors disagree with the reviewer's last point (phytoplankton biomass higher in 2016), as chlorophyll concentrations in both years were similar between years (within the same order of magnitude) and we would thus expect phytoplankton N uptake to be comparable (within the same order of magnitude).

Reviewer comment: I think the discussion at line 20 on pg 15 could also talk a little more about the system as a transformer of nitrogen importing DIN and exporting algal biomass as well as N derived from nitrogen fixation. At the moment it is a bit repetitive and not as interesting as it could be. I don't really think the term N pendulum is correct, it really modulates the inputs depending on residence time, with a net export of nitrogen from nitrogen fixation. This finding is consistent with a previous study of a shallow eutrophic lake which often showed net exports of total nitrogen, most likely due to nitrogen fixation. Cook, P.L.M., K.T. Aldridge, S. Lamontagne, and J.D. Brookes. (2010). Retention of nitrogen, phosphorus and silicon in a large semi-arid riverine lake system. Biogeo- chemistry, 99: 49-63. Author response: The second paragraph in section 4.4 will be expanded to include a more detailed discussion of the capacity of the system to transform inorganic forms of N to organic forms, thus modulating the magnitude and composition of N loading downstream to Lake Erie. The cited paper from the reviewer will lend support to this point. With regard to the term "pendulum," please refer to our response to comment #2 above.

Reviewer comment: The last paragraph of the discussion is quite speculative, I suggest remove. Author response: The final paragraph was meant to put our results in context within the larger Great Lakes system and the potential shifts in hydrology and nutrient regime associated with climate change. We prefer to keep this paragraph, shortening

it and emphasizing the potential application to future work rather than presenting a speculation.

Reviewer comment: Figure 2, micro symbol now appears as milli. Author response: There seems to have been a conversion error that made the micro symbol appear as "m." Thank you for pointing this out, and it will be fixed in the revised manuscript.

Reviewer comment: Figure 3 micro symbol as above Author response: There seems to have been a conversion error that made the micro symbol appear as "m." Thank you for pointing this out, and it will be fixed in the revised manuscript.

Reviewer comment: Figure 4 the letters showing statistically significant groupings are unclear. ËĘ is carat, not carrot Author response: The groupings in Figure 4 are the result of an unusual outcome of the Tukey's post-hoc test of the two-way interaction effects ANOVA, which are valid yet confusing. As this statistical result is not a crucial outcome of the manuscript, the authors will consider an alternate illustration of the result that does not distract from the message. The misspelling of the ˆ symbol will be revised.

Reviewer comment: Figure 5a. Why are these rates reported volumetrically? They should be areal as for Fig 3. Author response: Volumetric rates will be converted to areal rates for the budget calculations.

---

## Author Response (AR1)

Response to Editorial Comments

Editor comment: Following the reviewers' suggestions and based to my own reading of your MS and responses to the comments, I believe your paper can be published in Biogeosciences after some significant revisions. The two referees have raised important questions that must carefully addressed in your revised MS. This includes a better description of methodology and the improvement of Nitrogen mass-balance calculations by the different biogeochemical processes you have measured. I also find necessary the addition of "a brief section to the manuscript on the metagenomic analysis of the phytoplankton community" in order to sustain in your MS the discussion on the relationship between N fixation and HABs. I was in general satisfied with most of your answers to the referees comments, and I recommend you incorporate all the necessary information in your revised MS. The two referees also expressed very consistent criticisms on the way you use of two expressions in your MS: the "Nitrogen pendulum" and the "freshwater estuary". I found these comments very relevant and ask you consider them more seriously.

I think the term "pendulum" must be explained more clearly, as (if I well understood) this is the first time the "Nitrogen pendulum" is described. A pendulum oscillates, swings, up and down and right and left… please be more precise explaining the pertinence of the metaphor and what processes make the pendulum move up and down. The use of this expression in the title of your MS is very abrupt and I don't think BG readers will understand immediately what you are referring to in the title. I suggest you remove the expression from the title and include it in the abstract together with a short definition of the concept.

I have much more concern with the use of the term "estuary" or "freshwater estuary", even if the NOAA has defined this last term, the most commonly used definition of an estuary remains the one Cameron and Pritchard (1963): "a semi-enclosed coastal body of water, which has free connection with the open sea, and within which seawater is measurably diluted with freshwater derived from land drainage". More recently, many studies have extended the definition of the estuary to the "tidal river" (and sometimes "freshwater estuarine zone"), as the upper estuarine region without salt intrusion, but influenced by the tide. In Sandusky Bay there is no salt intrusion and there is no tide. So this ecosystem is very far away from the definition of an "estuary". I can understand the need for a specific term to describe the mixing zone between a river and a lake as this is an ecosystem very specific properties. "river-lake mixing zone" is probably the most precise term. I can eventually accept "freshwater estuary" in your MS after a careful definition (this definition is in contradiction with that of Cameron and Pritchard), although I feel this expression must be used sparingly and carefully. In all cases the use of "estuary" alone is inappropriate and your results cannot be extrapolated to "estuarine systems" where the salinity gradient is always a major driving force. For instance in estuaries, salinity stress strongly impacts microbial communities; salinity also generates desorption processes that control the availability of phosphate for phytoplankton. As a consequence, P limitation processes as discussed in your MS will not follow classical trends occurring in estuaries. As an example in your abstract the statement "Estuarine systems such as Sandusky Bay are

**mediators of…" is inappropriate because Sandusky Bay is NOT an estuary. In order to avoid confusion, your MS must be consistent with these definitions; Readers must know immediately that your study concerns FRESHWATERS and not ESTUARINE WATERS.**

**I will be pleased to read a revised version of your MS, together with a short response to my editorial comments.**

**With best regards,**
**Gwenaël Abril, Biogeosciences associate Editor.**

Author response: Thank you for your instructive editorial comments on our manuscript. We appreciate your feedback as well as the reviewers' feedback, both of which have improved the revised manuscript greatly. We have responded to specific reviewer comments below. Regarding the specific points brought up in your editorial comments:

- Methodological revisions are detailed in our response to review #2. This includes clarifications of the calculations involved with the isotope pairing technique, changes to the calculations associated with the N budget, and addition of text to explain rationale for various approaches.
- N budget revisions are made in the methods, results, and discussion. Table 1 has been removed and replaced with a new figure (figure 8). The changes are aimed at improving the ease of comparison among various N sources and sinks, which are now illustrated as a function of discharge patterns. See details on these revisions below in our responses to reviewers #1 and #2.
- A section on metagenomics has been added in section 2.3, the results section, throughout the discussion, and in figure 6. The addition of these data has allowed us to demonstrate (1) that *Planktothrix* dominates the cyanobacterial community throughout the sampling season and (2) N fixers are present as a component of the cyanobacterial assemblage. See details on these revisions below in our responses to reviewers #1 and #2.
- Use of "N pendulum" has been removed from the manuscript. We opted to remove this term completely rather than more explicitly define the term, as multiple internal and external reviewers have expressed confusion over the term. *Importantly, we have changed the title in the manuscript, but we are unsure about how to change the title in the manuscript submission system.*
- Use of "estuary" has been removed in favor of "river-lake mixing zone" and "drowned river mouth." Usage of "estuary" to describe true estuarine systems has been retained. See details on these revisions below in our responses to reviewers #1 and #2.

We look forward to your evaluation. We much appreciate your time and efforts.
Thank you,
Kateri Salk

**Response to Review #1**

**Reviewer comment: issues in associating chlorophyll concentrations with HABs**
Author response: Previous studies have demonstrated a strong correlation between chlorophyll concentrations and phytoplankton biomass in this system, and *Planktothrix* generally dominates phytoplankton biomass (Davis et al. 2015, Conroy et al. 2017). This point is emphasized in our introduction of the study system (page 3, lines 5-7) and in our discussion of the use of chlorophyll concentrations as a proxy for phytoplankton biomass in this system (page 13, lines 5-6). We have added a section to the manuscript on metagenomic analysis of the cyanobacterial community, which demonstrates the dominance of *Planktothrix* as well as the presence of a community of N fixers (page 5, lines 1-16; page 10, lines 20-22; page 14, lines 1-3).

**Reviewer comment: use of the term "Great Lakes estuary" or "freshwater estuary" may not be appropriate**
Author response: Usage of the term "estuary" when describing systems in the Great Lakes have been changed to "drowned river mouth" (page 1, line 22; page 3, line 4, page 15, line 4), "river-lake mixing zone" (page 3, line 17; page 14, line 29), or "coastal wetland" (page 13, line 2). Instances of the term "estuary" to describe systems fitting the traditional definition of an estuary have been retained (page 11, line 31; page 12, line 13; page 14, line 29, page 15, line 33). When direct comparisons are made between Sandusky Bay and estuarine systems, these comparisons are made with respect to common characteristics including swings in discharge and N loading from the river (e.g., page 15, lines 21-24).

**Reviewer comment: p. 11 – denitrification relies on carbon supply as well as nitrate, potential dependence on C:nitrate ratios**
Author response: Given the high rates of primary productivity in this system and the high carbon content of the sediments (e.g., Ostrom et al. 2005), the consideration of C limitation on denitrification was not a major concern for this study. Additionally, a recent study showed that sediment C:N ratios were not significant predictors of denitrification rates across the Great Lakes and in Lake Erie (Small et al. 2016). Considerations of C supply are mentioned in the methods along with the assumptions of the IPT method (page 6, lines 25-26).

Ostrom et al. 2005. Evaluation of primary production in Lake Erie by multiple proxies. Oecologia 144: 115-124. doi: 10.1007/s00442-005-0032-5

Small et al. 2016. Large differences in potential denitrification and microbial communities across the Laurentian great lakes. Biogeochemistry 128: 353. doi: 10.1007/s10533-016-0212-x

**Reviewer comment: p. 12 – obvious comments in lines 6-12 on $N_2O$ production as byproduct of denitrification, but logic linking this to last sentence in paragraph 2 is not justified**
Author response: The paragraph on $N_2O$ emissions in section 4.2 has been removed. The relevant information from this paragraph has been added to the first paragraph of section 4.2 (page 12, lines 1-2, 5), where we emphasize that denitrification is efficient (very little $N_2O$ released compared to $N_2$), and that nearshore areas of the Great Lakes have the potential to be influential zones of N removal. This has cleared up the mismatch in logic the reviewer points out.

**Reviewer comment: Section 4.3 – Impact of remineralization on N budget needs to be discussed further, particularly the distinction between the gross and net rate of phytoplankton uptake (is uptake a sink?)**

Author response: The paragraph on remineralization in the discussion has been edited to include not only the implications of remineralization on rate measurements of $NH_4^+$ uptake (as was present before) but also the importance of remineralization in eutrophic systems that experience HABs and the significance of a rapidly cycling $NH_4^+$ pool despite challenges to quantify transient availability and uptake (page 13, lines 21-32). The concept of recycling as a mechanism for generating a supply of N under N limitation is also echoed later in section 4.3 (page 14, lines 16-19).

**Reviewer comment: Section 4.3 – These results encourage a thorough investigation of the controls of nitrogen fixation, including a supply of P and Fe**

Author response: The reviewer points out an interesting and relevant consideration, which the authors considered as well in the preparation of this manuscript. While Fe limitation of N fixation has precedent in the literature in several systems, we dismissed this mechanism for two reasons: (1) as the Sandusky River discharges into the bay, it brings a high sediment load that is expected to have a sufficient trace metal load to support phytoplankton demands, and (2) the sediments in Sandusky Bay are anoxic, and it is likely that nighttime sediment oxygen demand is high enough that transient anoxia develops at the sediment-water interface, thus enabling P and reduced Fe to be released into the water column and be taken up by phytoplankton. These internal loading events, while outside the scope of this study, have been captured on several occasions over the course of multi-year surveys of the bay. While P limitation was certainly at play during portions of the season in both years, the substantial phytoplankton biomass and high N fixation rates indicated that P was likely not a driving factor for the activity of N fixation. We have added a section to the manuscript on metagenomic analysis of the cyanobacterial community, which demonstrates the presence of a community of N fixers, which lends additional support to the biogeochemical measurements of N fixation (page 5, lines 1-16; page 10, lines 20-22; page 14, lines 1-3).

**Reviewer comment: Section 4.4, line 29 – not sure it is straightforward to extrapolate this argument to other coastal areas (likely that water flow is not operating as a pendulum in other coastal systems, where tides and other factors might be stronger controls)**

Author response: The final section of the manuscript has been revised with a greater emphasis on Sandusky Bay and the implications of future changes on the role of this system in the Great Lakes system. Extensions to other coastal areas (i.e., Gulf of Mexico) have been retained, as the specific systems referenced have been well-documented as systems that are highly influenced by riverine N loading. We do not suggest that this is the only factor to consider, simply that many systems are highly influenced by hydrology and nutrient loading, so this study could lend insight into other systems (page 15-16, lines 26-4).

**Response to Review #2**

**Reviewer comment: Abstract I suggest referring to dissimilatory nitrate reductive processes as opposed to assimilation**
Author response: Assimilatory (phytoplankton uptake) and dissimilatory (microbial N reduction) processes are considered separately, and this language has been revised to more clearly reflect this distinction (page 1, line 15).

**Reviewer comment: Use of 'pendulums' not really the right term in my opinion. As elaborated on below, I think the key point is that the system is a modulator of nutrient inputs.**
Author response: The use of the term "pendulum" has been removed from the manuscript. In cases where swings in hydrology and N dynamics are pointed out, we have chosen to use terms such as "oscillation" and "modulate" as descriptors for how the system functions.

**Reviewer comment: Pg 3 line 5 – I don't think estuary is generally accepted as a term for rivers entering freshwater lakes – I suggest mixing zone.**
Author response: Usage of the term "estuary" when describing systems in the Great Lakes have been changed to "drowned river mouth" (page 1, line 22; page 3, line 4, page 15, line 4), "river-lake mixing zone" (page 3, line 17; page 14, line 29), or "coastal wetland" (page 13, line 2). Instances of the term "estuary" to describe systems fitting the traditional definition of an estuary have been retained (page 11, line 31; page 12, line 13; page 14, line 29, page 15, line 33). When direct comparisons are made between Sandusky Bay and estuarine systems, these comparisons are made with respect to common characteristics including swings in discharge and N loading from the river (e.g., page 15, lines 21-24).

**Reviewer comment: Methods Isotope analysis delta15N values are mentioned in the methods, why? The isotope pairing equations use excess ratios of M/Z 29/28 and 30/28 for N2 and 45/44 and 46/46 for N2. I suggest deleting all ref to del 15N and explaining which masses were monitored and how excess ratios were calculated. It also not clear why N2 was also measured with MIMS or how these data were used.**
Author response: The equation for delta (equation 1 in previous version) was removed (as the reviewer points out, we are working not with deltas but with isotope fractions). A description of how $P_{29}$, $P_{30}$, $P_{45}$, and $P_{46}$ were calculated was added, detailing that the change in isotope fraction of the relevant mass was measured by IRMS and the calculation was corrected for the change in concentration of $N_2$ or $N_2O$ as measured by MIMS or GC-ECD, respectively (page 6, lines 31-33; page 7, lines 2-4).

**Reviewer comment: Phytoplankton N uptake 15NH4/NO3 contamination of 15N2. You state that uptake of contamination would have made up less than 5% of measured rates. This depends on the rates. Is this even the case for the lowest measured rates? The main thing that convinced me your data were probably ok, was the fact you could measure low rates in 2016.**
Author response: The 5% value for potential contamination is the maximum potential contamination across all measurements, from low to high rates. We calculated the possible overestimation of N fixation given the level of potential contamination reported in Dabundo et

al. (2014), and 5 % was the highest calculated value across the dataset. This statement was revised to clarify this point (page 7, lines 20-22).

**Reviewer comment: Budget I don't think converting sediment process rates to volumetric rates is meaningful – these should either be shown as areal rates or total mass for the whole system.**
Author response: Sediment process rates have been changed back to areal rates in figure 7, consistent with the units presented in figure 4. Water column rates have been retained as volumetric rates in figures 5 and 7. For calculations of budget, all rates have been converted to kg/d (from either areal or volumetric rates), enabling direct comparisons. The methodology for these conversions is detailed in section 2.6 (page 8, lines 16-19).

**Reviewer comment: Line 5 pg 8 – I agree with your point about TKN in the river, but what about in Sandusky Bay? As mentioned below, I think the system is really a modulator that converts NOx to organic matter and this will be shown clearly in the TKN data if available.**
Author response:
TKN concentrations in the Sandusky River and Sandusky Bay were added to the paper in the methods (page 3, line 31-32; page 4, line 8, 22-23; page 8, lines 19-20), results (page 9, line 13-14; page 10, lines 32-33), and figures 3 and 8. This includes an examination of concentrations as well as how TKN loads compare to other aspects of the N budget. The observation of higher TKN concentrations in the bay vs. the river suggests a conversion of DIN into TKN (recycling to organic forms or ammonium), which the reviewer points out is an indication that the system converts nitrate into organic matter and ammonium. We point out the indication of recycling in section 4.3 (page 13, lines 24-26; page 14, lines 17-19).

**Reviewer comment: For the nitrate loads, at what time interval were concentration and flow measured? How were these data interpolated to calculate loads?**
Author response: Daily nitrate loads were calculated based on daily discharge and nitrate concentration data from the Heidelberg NCWQR. Direct comparisons were made for days when assimilatory and dissimilatory N transformations were measured. These calculations are detailed in section 2.6 (page 8, lines 19-20) and in figure 8.

**Reviewer comment: Discussion N removal processes. This was generally good and I agree with the arguments. I felt, however this section could have been a little more quantitative. For example, it is argued that the increase in the N:P was driven in a large part by denitrification. I suggest the authors undertake a back of the envelope calculation to show how the change in the mass of NO3 in the water column over this period of NO3-drawdown compares with total denitrification measured over the same period.**
Author response: The statement about short hydraulic residence times reducing the capacity for denitrification to remove significant amounts of N has been amended to be more quantitative (page 12, lines 24-26). Given the covarying rates of varying processes over time (N loading, phytoplankton uptake, denitrification) as well as the complex hydrology, we have opted not to make a calculation of the comparison between changing water column $NO_3^-$ concentrations and denitrification rates. We feel it is misleading to make a back-of-the-envelope calculation here, as the assumptions we would need to make about hydraulic residence time, N recycling, and

temporal variation in rates would be an oversimplification of this system. Instead, we have focused our efforts on illustrating the N budget as a function of river discharge in figure 8.

**Reviewer comment: Discussion N removal processes. Also of relevance here is that Dw (water column driven denitrification) and Dn (water column driven denitrification) are not reported. The breakdown of these is important when considering the drawdown rate of NOx.**
Author response: Given the shallow and well-mixed nature of this system (mean depth 1.6-2.6 m), water column denitrification was not considered as an appreciable denitrification source. This assumption was borne out in the IPT calculations, which enable distinction between Dw and Dn and confirmed that Dw was not active. These detailed results were not included in the manuscript, but a statement describing the assumption of negligible water column denitrification was added (page 4, lines 15-16).

**Reviewer comment: Budget I think there was a missed opportunity with the budget to integrate the findings a little more clearly. I suggest that for each period process rates were measured, a budget be undertaken (could be daily or perhaps monthly basis). These budget terms could then graphed to highlight the change from high catchment inputs to high internal inputs via N fixation as flows decreased through to August. This would also highlight the relatively minor importance of denitrification as a sink compared to the inputs. Although the phytoplankton assimilation measurements are a nice part of the paper, I don't think they can be used meaningfully in the budget because they were taken in 2016 when phytoplankton biomass was higher.**
Author response: The reviewer brings up a good point that the budget calculations could be more effectively integrated to illustrate the system more clearly. Table 1 has been removed and instead we have added figure 8, which shows all the measured aspects of the N budget as (1) total ranges, (2) individual rate measurements, and (3) as a function of discharge. This figure enables direct comparisons between processes and examinations of how river discharge impacts the system. A description of the budget is detailed in the results (page 10-11, lines 32-4) and throughout the discussion.

**Reviewer comment: I think the discussion at line 20 on pg 15 could also talk a little more about the system as a transformer of nitrogen importing DIN and exporting algal biomass as well as N derived from nitrogen fixation. At the moment it is a bit repetitive and not as interesting as it could be. I don't really think the term N pendulum is correct, it really modulates the inputs depending on residence time, with a net export of nitrogen from nitrogen fixation. This finding is consistent with a previous study of a shallow eutrophic lake which often showed net exports of total nitrogen, most likely due to nitrogen fixation. Cook, P.L.M., K.T. Aldridge, S. Lamontagne, and J.D. Brookes. (2010). Retention of nitrogen, phosphorus and silicon in a large semi-arid riverine lake system. Biogeo-chemistry, 99: 49-63.**
Author response: The second paragraph in section 4.4 was revised to include a more detailed discussion of the capacity of the system to transform inorganic forms of N to organic forms, thus modulating the magnitude and composition of N loading downstream to Lake Erie (page 15, lines 16-24). Leading up to this paragraph, supporting points were revised and added (page 13, lines 21-32; page 14, lines 16-19; page 15, lines 9-10). We have cited the suggested study in

section 4.4 as well (page 15, lines 8-9). We have removed the use of the term "pendulum" in favor of terms such as "modulator" and "oscillations."

**Reviewer comment: The last paragraph of the discussion is quite speculative, I suggest remove.**
Author response: The final paragraph is meant to put our results in context within the larger Great Lakes system and the potential shifts in hydrology and nutrient regime associated with climate change. We have kept this paragraph in the manuscript, but we have removed two sentences and revised another sentence to limit speculation (page 15-16, lines 26-4).

**Reviewer comment: Figure 2, micro symbol now appears as milli.**
Author response: There seems to have been a conversion error that made the micro symbol appear as "m." Thank you for pointing this out, and it is fixed in the revised manuscript.

**Reviewer comment: Figure 3 micro symbol as above**
Author response: There seems to have been a conversion error that made the micro symbol appear as "m." Thank you for pointing this out, and it is fixed in the revised manuscript.

**Reviewer comment: Figure 4 the letters showing statistically significant groupings are unclear. ˆ is carat, not carrot**
Author response: The groupings in Figure 5 (formerly figure 4) are the result of an unusual outcome of the Tukey's post-hoc test of the two-way interaction effects ANOVA, which are valid yet confusing. As this statistical result is not a crucial outcome of the manuscript, the authors have removed the letters from the figure. The misspelling of the ^ symbol is revised.

**Reviewer comment: Figure 5a. Why are these rates reported volumetrically? They should be areal as for Fig 3.**
Author response: Figure 7a (formerly figure 5a) is now presented in areal units, consistent with rates presented in figure 4.

[revised manuscript text omitted]

**2.3. 16S rRNA Metagenomic Analysis**

DNA was extracted using the PowerWater Sterivex DNA Isolation Kit (MO BIO Laboratories, Inc, Carlsbad, CA, USA) following manufacturer's instructions. Short 16S rRNA Illumina amplicon tag (iTag) sequencing of the V4-V5 hypervariable region of bacterial genomes was completed at the Joint Genome Institute (JGI; Walnut Creek, CA, USA) using an Illumina MiSeq benchtop sequencer ($2 \times 301$ bp reads) according to standard JGI procedures (Tremblay et al., 2015). Primer design for universal amplification of the V4-V5 region of 16S rDNA was based on Parada et al. (2016). Resulting sequences were demultiplexed and contaminating Illumina adaptor sequences were removed using the kmer filter in BBDuk (v37.62) following Singer et al. (2016). Briefly, BBDuk was used to remove reads containing more than 1 ′N′ base, or with a quality score < 10 across the read, or length $\leq$ 51 bp or 33% of the full length read. Additional processing using BBMap (http:\\bbtools.jgi.doe.gov) mapped reads to masked human, cat, dog and mouse references, discarding hits exceeding 93% identity (Singer et al., 2016).

Processing, clustering and classification of sequenced reads was performed as described previously (Tremblay et al., 2015). Briefly, quality controlled reads were processed by iTagger (v2.2), first by read clustering (97% identity) using algorithms in USEARCH (v9.2 ; Edgar, 2010), followed by the assignment of operational taxonomic units (OTUs) using the SILVA 16S SSU database (v128; Quast et al., 2013), and finally, analysis of ecological data using QIIME v1.9.1 (Caporaso et al., 2010).

**2.4 Sediment Microbial N Removal**

Upon return to the lab, water from station 1163 was gently added to cores to a depth of 20 cm. Cores were pre-incubated for 12 h in the dark at *in situ* temperature under gentle aeration to maintain oxic conditions in the overlying water. Following pre-incubation of sediment cores, a sample was taken from the overlying water for DIN concentration analysis ($NO_3^-$, $NO_2^-$ and $NH_4^+$), filtered through a precombusted GF/F filter and frozen until analysis. $^{15}N$-$NO_3^-$ (100 μmol $L^{-1}$) was then added to the overlying water in each core. Cores were then capped and statically incubated under gentle stirring throughout the duration of the experiment. An initial equilibration period was employed to allow homogenization of $NO_3^-$ between the overlying water and the $NO_3^-$ reduction zone in the sediment porewater (Dalsgaard et al., 2000). Cores were sacrificed in triplicate or quadruplicate at intervals of 0, 3 or 4, and 6 h, during which time oxic conditions were maintained in the overlying water. Dissolved $O_2$ in the overlying water was monitored to evaluate the maintenance of oxic conditions throughout the incubation using a YSI 600QS sonde. A final sample for DIN analysis was collected when each core was sacrificed and processed as described above.

Samples for the determination of $\delta^{15}N_2$ were collected according to Hamilton and Ostrom (2007); briefly, dissolved gases were equilibrated with a He atmosphere, and the headspace was transferred into a pre-evacuated 12 mL Exetainer (Labco Ltd, Lampeter, Ceredigion, UK). Samples for analysis of dissolved $N_2$ concentrations were siphoned into 12 mL Exetainers to

overflowing and amended with 200 µL of saturated ZnCl₂ solution to halt biological activity. All Exetainers were stored underwater at room temperature to minimize diffusion of atmospheric N₂ during storage. Samples for analysis of the $\delta^{15}$N₂O and N₂O concentrations were siphoned into 250 and 60 mL serum bottles, respectively, to overflowing and sealed without a headspace with a butyl rubber septum. Biological activity was halted by adding saturated HgCl₂ solution to a final

5   concentration of 0.4 % by volume.

Prior to N₂O concentration analysis, a headspace of 20 mL He was introduced in each 60 mL bottle, maintaining atmospheric pressure with a vent needle. Serum bottles were allowed to equilibrate under gentle shaking for at least 12 h prior to analysis. The headspace was then analyzed by GC-ECD (Shimadzu Greenhouse Gas Analyzer GC-2014, Shimadzu Scientific

10   Instruments, Columbia, MD, USA) for N₂O concentration. The dissolved concentration was calculated based on the headspace equilibrium concentration (Hamilton and Ostrom, 2007).

The isotopic composition of N₂O was analyzed upon introduction of sample water into an enclosed 0.75 L glass vessel that was previously purged of atmospheric air using a gentle flow of He. Dissolved gases were subsequently stripped from the

15   water by sparging the sample with He (Sansone et al., 1997), which carried sample gases into a Trace Gas sample introduction system interfaced to an Isoprime isotope ratio mass spectrometer (IRMS; Elementar Americas, Inc., Mount Laurel, NJ, USA).

Concentrations of dissolved N₂ were analyzed by membrane inlet mass spectrometry (MIMS; Kana et al., 1994). The isotopic composition of N₂ was analyzed by introducing the sample to an evacuated 800 µL sampling loop and then onto a packed

20   molecular sieve (5 Å) column (Alltech, Inc., Deerfield, IL) using He carrier gas within a gas chromatograph (HP-5980, Hewlett Packard, Ramsey, MN) interfaced to an Isoprime IRMS. Analytical reproducibility of standards was 0.3 ‰.

Denitrification, anammox, and N₂O production rates were calculated by the IPT. Calculations were derived from the IPT$_{anaN2O}$ (Hsu and Kao, 2013), which builds on the R-IPT (Risgaard-Petersen et al., 2003) by enabling quantification of N₂O production

25   simultaneously with denitrification and anammox. This approach relies on the assumption that N removal is limited by N and not by other factors such as C, an assumption that has been supported for Sandusky Bay by Small et al. (2016). Briefly, N₂ production by denitrification ($D_{14\text{-}N_2}$) and anammox ($A_{14}$) were calculated as:

$$D_{14\text{-}N_2} = (r_{14\text{-}N_2O}+1) \times 2r_{14\text{-}N_2O} \times P_{30} \qquad\qquad (1)$$

and

30   $$A_{14} = 2r_{14\text{-}N_2O} \times (P_{29}-2r_{14\text{-}N_2O} \times P_{30}) \qquad\qquad (2)$$

where $r_{14\text{-}N_2O}$ is the ratio of $^{14}$N to $^{15}$N in N₂O, $P_{29}$ and $P_{30}$, the production of $^{29}$N₂ and $^{30}$N₂, respectively, were calculated by the increase in isotope fraction of mass 29 or 30 as measured by IRMS and corrected for the change in total N₂ concentration as measured by MIMS. N₂O production ($D_{14\text{-}N2O}$) was calculated as:

$\delta = \frac{R_{sam} - R_{std}}{R_{std}} \times 1000$ ++++++++++++ (1)¶
where R$_{sam}$ is the isotope ratio of the sample, R$_{std}$ is the isotope ratio of the standard, and $\delta$ is reported as per mil (‰).

$$D_{14\text{-}N_2O} = r_{14\text{-}N_2O} \times (2P_{46} + P_{45})$$ (3)

where $P_{45}$ and $P_{46}$ are the production of $^{45}N_2O$ and $^{46}N_2O$, respectively. $P_{45}$ and $P_{46}$ were calculated by the increase in isotope fraction of mass 45 or 46 as measured by IRMS and corrected for the change in total $N_2O$ concentration as measured by GC-ECD.

5 **2.5 Phytoplankton N Uptake**

[revised manuscript text omitted]

Metagenomic analysis revealed that *Planktothrix* spp. dominated the cyanobacterial community (22-99 % of iTag reads) on all but one sampled date at station 1163. Diazotrophs (*Aphanizomenon* sp. and *Dolichospermum* spp.) were present on all sampled dates and made up a minor assemblage of the cyanobacterial community (1-33 % of iTag reads; Fig. 6).

DIN concentration was positively correlated with daily volumetric rates of denitrification, $N_2O$ production, and $NO_3^-$ uptake, explaining 64, 60, and 75 % of variance in mean rates, respectively (Fig. 7). Anammox rates were not correlated with DIN concentration due to the high variance observed among dates and replicate sediment cores (Fig. 7b). DIN concentration explained only 12 % of variance in mean $NH_4^+$ uptake rates, although $NH_4^+$ concentration alone explained 86 % of variance and was positively correlated with $NH_4^+$ uptake. N fixation was not significantly correlated with DIN concentration (Fig. 7a). Ranges in daily volumetric rates of assimilatory processes ($NO_3^-$ uptake, $NH_4^+$ uptake, N fixation) were several orders of magnitude higher than dissimilatory processes (denitrification, anammox, $N_2O$ production), with the exception of $NH_4^+$ uptake and denitrification, which were within the same magnitude (Fig. 7).

Daily rates of $NO_3^-$ and TKN loading from the Sandusky River varied by five and two orders of magnitude, respectively, and high loading rates were associated with high discharge (Fig. 8). N fixation often exceeded riverine N loading as a source of N

to Sandusky Bay, particularly during periods of low discharge. NO$_3^-$ uptake was the dominant N uptake process in Sandusky Bay, outpacing NH$_4^+$ uptake and dissimilatory N removal processes (Fig. 8). On the basis of the total magnitude of ranges, sources and demands of N in this system are tipped in favor of a net source to Lake Erie, ranging from a strong source during periods of high discharge to a weak source during periods of low discharge.

**4 Discussion**

**4.1 Nutrient Stoichiometry**

Sandusky Bay displays considerable seasonal variation in nutrient concentrations, molar dissolved N:P ratios, and chl *a* concentrations, indicative of a system with dynamic changes in hydrology and biogeochemical activity. Maximum chl *a* concentrations in both years (> 100 µg L$^{-1}$) were similar to other hypereutrophic systems (Zhang et al., 2011; Wheeler et al., 2012; Steffen et al., 2014b), as were large swings in NO$_3^-$ concentrations in 2015 (Xu et al., 2010; Steffen et al., 2014b; McCarthy et al., 2016). Elevated nutrient concentrations were associated with high discharge events from the Sandusky River, demonstrating a strong watershed influence on Sandusky Bay. Indeed, the Sandusky River watershed comprises an area 30 times larger than the bay and delivers large nonpoint loads of N and P to its receiving waters (Robertson and Saad, 2011). Between the two study years, discharge from the Sandusky River varied substantially (tenfold higher in 2015), exhibiting large inter-annual variability in hydraulic residence time and nutrient concentrations.

As discharge from the Sandusky River decreased throughout the summer in both years, dissolved N:P ratios fell from a maximum of over 10,000 to below the threshold for N limitation. The decline in N:P ratios is largely driven by decreases in NO$_3^-$ concentration, particularly in 2015, as the range in PO$_4^{3-}$ and NH$_4^+$ concentration was comparatively narrow. Consumption of NO$_3^-$ could be attributed to both assimilatory and dissimilatory NO$_3^-$ reduction. If phytoplankton were solely responsible for the decline in NO$_3^-$, nutrients would be expected to be drawn down in molar proportions of approximately 16N:1P (Sterner and Elser, 2002). However, N:P ratios in Sandusky Bay fall sharply throughout the summer, while PO$_4^{3-}$ concentrations are relatively constant by comparison. Although this trend could be influenced by luxury uptake of N by phytoplankton and internal P loading from sediments (Filbrun et al., 2013; McCarthy et al., 2016), the dramatic depletion in DIN compels consideration of microbial N removal processes as a major mechanism for N drawdown in Sandusky Bay.

**4.2 N Removal Processes**

Marked declines in N:P with time and occurrence of N:P ratios < 16 provide compelling evidence that microbial N removal processes (i.e., denitrification and/or anammox) consume appreciable quantities of NO$_3^-$ in Sandusky Bay. Sediment $^{15}$N tracer incubations indicate the primary N removal mechanism in Sandusky Bay is denitrification, which comprised an average of 84 % of sediment N removal across sampling dates. Denitrification rates were positively correlated with DIN concentration, consistent with observations that N supply controls sediment denitrification capacity in estuaries, lakes, and continental shelves

(Seitzinger et al., 2006). Sandusky Bay exhibited efficient denitrification, with an average of only 2 % of sediment N removal released as $N_2O$. Denitrification rates in Sandusky Bay (10.02-64.81 $\mu$mol N $m^{-2}$ $h^{-1}$) are among the highest reported for the Laurentian Great Lakes. Previous denitrification measurements in offshore zones of the Great Lakes vary over four orders of magnitude, with western and central Lake Erie exhibiting the highest rates at $51 \pm 41$ $\mu$mol N $m^{-2}$ $h^{-1}$ (Small et al., 2014; 2016).

5   Nearshore zones, bays, and river mouths have been observed as areas of enhanced denitrification and $N_2O$ production compared to offshore zones (McCarthy et al., 2007; Small et al., 2014; 2016; Salk et al. 2016), reinforcing that Sandusky Bay and other shallow coastal areas have the potential to act as hotspots of N removal in the Great Lakes system.

Anammox activity in Sandusky Bay was highly variable, even among replicate sediment cores from the same site and date.
10   High variability was also observed in a study of potential anammox rates in the water column of Sandusky Bay and Lake Erie (Lu et al. 2018). Marked variability may be characteristic of anammox activity in freshwater environments, even across small spatial and temporal scales (Yoshinaga et al., 2011; Zhu et al., 2013; 2015). Anammox made up an average of 14 % of sediment N removal across the sampling period, indicating anammox activity in Sandusky Bay may be typical of shallow estuarine and freshwater systems (Thamdrup and Dalsgaard, 2002; Dalsgaard et al., 2005; Schubert et al., 2006; Dong et al., 2009; 2011;
15   Hsu and Kao, 2013; McCarthy et al., 2016). Measurements of active anammox via isotope tracers are consistent with detection of 16s RNA associated with anammox taxa in the sediment of Lake Erie (Small et al., 2016), demonstrating that anammox has the potential to be an appreciable N removal pathway in this and other nearshore regions within the Great Lakes.

[revised manuscript text omitted]

5 Bay is a source of N to Lake Erie, both as a conduit of watershed N loading and through introductions of fixed N. The magnitude of N delivered downstream is highly dependent on Sandusky River discharge (Fig. 2, 8). During portions of the year when riverine DIN loading is low, N fixation supplements DIN loading to meet assimilatory and dissimilatory N demands, representing a large and crucial balance for the N budget in Sandusky Bay. N fixation has been suggested as a mechanism for balancing the N budget in favor of a net source in other eutrophic lakes (Cook et al., 2010). N fixation and phytoplankton DIN uptake transfer N into organic forms that can be recycled within the system or delivered downstream. The dominance of

10 assimilatory processes suggests that although DIN concentrations are often low in Sandusky Bay, there is an actively cycling N stock within the phytoplankton community that may be utilized by *Planktothrix*. Dissimilatory sinks, although on a smaller magnitude, represent a permanent N sink that may have a greater influence on the development of N limitation than assimilatory processes.

The mass balance of N in Sandusky Bay undergoes rapid and dramatic seasonal transitions, shifting the role of the bay from a strong to a weak source of N to Lake Erie. Previous work suggests that oscillations between excess N abundance to N limitation that are consistent from year to year (Conroy et al., 2007; Davis et al., 2015; Conroy et al., 2017). During periods of high discharge and N loading, short hydraulic residence times prevent substantial processing of N and DIN is flushed into Lake

20 Erie. When Sandusky River discharge and N loading are low, sediment N removal and N recycling by phytoplankton result in a smaller pulse of N delivery to Lake Erie that has been extensively processed into organic forms. Sandusky Bay thus oscillates between acting as a conduit and a filter of N. Discharge-driven oscillations in N cycling and downstream delivery may be a common feature in river-lake mixing zones and estuaries, most notably Narragansett Bay (Fulweiler et al., 2007; Fulweiler and Heiss, 2014).

Future projections suggest that climate change will create conditions that are likely to intensify the role of Sandusky Bay and comparable systems as conduits of downstream nutrient delivery. Overall precipitation in the watershed is expected to increase and become dominated by incidences of extreme precipitation (Prein et al., 2017). Moreover, increases in precipitation are predicted to be accompanied of enhanced riverine N loading in the near future (Sinha et al., 2017). Increases in the frequency

30 and intensity of discharge and riverine N loading will likely diminish the capacity for Sandusky Bay to transform and remove N, thus favoring the downstream export of DIN. The increased export of nutrients downstream may support cHABs in the central basin of Lake Erie and exacerbate water quality issues downstream, including hypoxia in the central basin of Lake Erie and the St. Lawrence Estuary (Lehmann et al., 2009; Michalak et al., 2013). Similar climate change-driven shifts in water

[revised manuscript text omitted]

**Tables**

Table 1. Total range in N sources, assimilatory N uptake processes, and dissimilatory N sinks in Sandusky Bay.

| Process | Minimum (kg d$^{-1}$) | Maximum (kg d$^{-1}$) |
|---|---|---|
| **Sources** | | |
| DIN loading | 1 | 355,054 |
| N fixation | 2,143 | 102,179 |
| *Total* | *2,144* | *457,233* |
| | | |
| **Assimilatory N uptake** | | |
| NH$_4^+$ uptake | 267 | 6,264 |
| NO$_3^-$ uptake | 1,360 | 104,709 |
| *Total* | *1,627* | *110,973* |
| | | |
| **Dissimilatory N uptake** | | |
| Denitrification | 274 | 1,769 |
| Anammox | 14 | 221 |
| N$_2$O production | 2 | 64 |
| *Total* | *290* | *2,054* |